# What Happens Next? Anticipating Future Motion by Generating Point Trajectories

**Gabrijel Boduljak**✉  **Laurynas Karazija**  **Iro Laina**  **Christian Rupprecht**  **Andrea Vedaldi**

Visual Geometry Group, University of Oxford

## Abstract

We consider the problem of *forecasting motion* from a single image, i.e., predicting how objects in the world are likely to move, without the ability to observe other parameters such as the object velocities or the forces applied to them. We formulate this task as conditional generation of dense trajectory grids with a model that closely follows the architecture of modern video generators but outputs motion trajectories instead of pixels. This approach captures scene-wide dynamics and uncertainty, yielding more accurate and diverse predictions than prior regressors and generators. Although recent state-of-the-art video generators are often regarded as world models, we show that they struggle with forecasting motion from a single image, even in simple physical scenarios such as falling blocks or mechanical object interactions, despite fine-tuning on such data. We show that this limitation arises from the overhead of generating pixels rather than directly modeling motion.

## 1 Introduction

We consider the problem of *forecasting motion* from a single image, i.e., predicting how objects in the world are likely to move. This task is representative of an agent trying to infer what may happen next given only its limited observations of the environment. Because a single image does not fully specify the observed physical system, many different futures are possible and must be predicted as potential outcomes. Yet, these predictions are not arbitrary: they must be consistent with the image, physical principles, and facts about the observed objects that are known *a priori*.

Modeling such a prior is important in many applications of AI, such as generating realistic videos, policy learning (Wen et al., 2024a; Yang et al., 2025; Bharadhwaj et al., 2024b), model-based control (Ding et al., 2024; Mazzaglia et al., 2024; Yang et al., 2024c;b;a), and other problems that require an understanding of physical phenomena.

As others before us, particularly in robotics (Wen et al., 2024a; Yang et al., 2025), we formulate motion forecasting as predicting the trajectories of points in the input image. However, unlike prior work, we formulate this problem as *generating* the trajectories conditioned on the observed image. This stochastic formulation is more appropriate as it can model the forecasting ambiguity as a *distribution* over possible futures, which can then be sampled to produce likely realizations.

Increasingly powerful video generators (Polyak et al., 2025; HaCohen et al., 2024b; Wan et al., 2025; Brooks et al., 2024; Parker-Holder et al., 2024; NVIDIA et al., 2025) address a similar forecasting problem, predicting a video starting from a single image. We thus suggest making our formulation similar to many such video generators, and in particular, we use flow matching (Liu et al., 2023b; Lipman et al., 2022). However, instead of generating pixels, we generate their trajectories on a grid.

Previous motion forecasters (Wen et al., 2024a; Yang et al., 2025; Bharadhwaj et al., 2024b) generally focus on predicting the motion of selected image points, for example, those that land on the arm of a robot. In contrast, our formulation, inspired by video generation, is (quasi-)dense: we predict the motion of all points in a grid. This allows the model to reason about the entire scene jointly (Karaev et al., 2024b). This is beneficial because, as time passes, objects that may be too far apart to interact initially may eventually collide.

---

✉Correspondence to: gabrijel@robots.ox.ac.uk

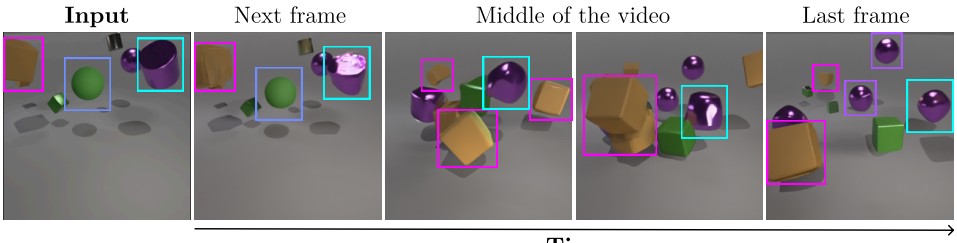

Figure 1: **State-of-the-art video generators frequently produce unrealistic motion.** Even state-of-the-art models, such as WAN 14B shown here, often struggle to produce accurate, realistic and coherent motion. Common failure modes include *distorted object geometry*, *objects splitting into multiple parts*, *objects disappearing*, and *objects spontaneously appearing* throughout the video.

We also ask whether video generators can do more than provide a good architecture. Many have suggested that training video generators on billions of videos is an effective way of learning *world models* (Ha & Schmidhuber, 2018). If so, we could *reduce* motion forecasting to video generation, for example, by applying a point tracker to the generated video (Bharadhwaj et al., 2024a).

Even so, we hypothesize that predicting point trajectories directly can be significantly simpler and more efficient than predicting pixels followed by inferring motion from the generated video. Trajectories capture motion directly, whereas videos need to be further translated into an estimate of motion (Ko et al., 2023; Patel et al., 2025). Using trajectories also injects two inductive biases, *object permanence* and *temporal coherence*, that general video generators struggle with (Motamed et al., 2025; Kang et al., 2024) (Figure 1).

To better contrast video and track generators, we minimize the difference between architectures and compare a trajectory forecaster trained from scratch to video generators trained on billions of videos. We show that even state-of-the-art video models like WAN (Wan et al., 2025) struggle with predicting the motion of objects in relatively simple simulated scenarios (Figure 1) *even after fine-tuning* them on these domains. Hence, it is unclear whether training video models on billions of general-purpose videos allows them to learn basic physical consistency. In contrast, learning to predict tracks is much more effective in this respect, even without pre-training on massive datasets.

For evaluation, we primarily consider synthetic scenarios, where it is possible to simulate different events starting from the same initial image. This facilitates assessing the ability of a model to capture the distribution of possible futures. To do so, we further adopt the *motion distributional metrics* from the video generation literature (Brooks et al., 2024). However, these population metrics are coarse and may not capture well the physical plausibility of the predicted motion. We thus consider scenarios where we can also test directly for aspects of physical consistency, such as maintaining the shape of rigid objects.

We experiment using simulated physical scenarios from Kubric (Greff et al., 2022), LIBERO Liu et al. (2023a), and Physion (Bear et al., 2022). We also consider the real-world setting using Physics101 (Wu et al., 2016). In all these cases, we show the advantage of our formulation compared to previous point track forecasters as well as video generators.

To summarize, our contributions are as follows:

1. We formulate the problem of image-based motion forecasting as generating the trajectories of a grid of image points, utilizing an architecture inspired by popular video generators.
2. We show that our model outperforms previous point track forecasters because it uses generation instead of regression, which models uncertainty, and considers points across the scene instead of focusing on a small subset of active points, which captures context better.
3. We further compare learning trajectory predictors from scratch to using state-of-the-art video generators pre-trained on billions of videos (further fine-tuned on our experimental domain) and show that the former can learn motion more efficiently and accurately.
4. We evaluate models using several different synthetic and real scenarios, and include metrics that test directly for certain aspects of physical plausibility such as rigidity.

## 2 RELATED WORK

**Visual motion forecasting.**   We consider the problem of forecasting possible motions of objects, expressed as points, in a scene, given a single image. Several variants of this task have been studied in the literature. Among the most relevant are Any-point Trajectory Modeling (ATM) (Wen et al., 2024b), Yang et al. (2025) Tra-MoE (Yang et al., 2025) and Track2Act (Bharadhwaj et al., 2024b) which focus on robotic control. ATM uses CoTracker (Karaev et al., 2024b) to pseudo-label the LIBERO dataset (Liu et al., 2023a) by tracking the motion of a robotic arm performing object manipulation tasks. Tra-MoE improves ATM with a mixture of experts. Similarly, Track2Act pseudo-labels both real-world action (Goyal et al., 2017; Damen et al., 2022) and robotics datasets (Brohan et al., 2022; Walke et al., 2023). ATM and Tra-MoE deterministically regress 32 points, while Track2Act generates 400 trajectories using diffusion (Ho et al., 2020). However, these methods focus only on active points, placed on the robot actuator and targets, and condition their predictions on a goal (expressed as a goal image in Track2Act). In contrast, we predict trajectories for many more points, placing them uniformly on a grid, and forecast them independently of whether they should be static or dynamic. This way, we cover the whole scene, modeling motion arising from the properties of the world. Recently, Pandey et al. (2024) introduced a training-free method that leverages Motion-I2V (Shi et al., 2024), a pre-trained image-to-flow model, to discover potential object motion within a given image. The method uses hand-crafted energy functions to guide the image-to-flow generator, aiming to separate object and camera movement. However, it can only handle a single, pre-segmented object, and it is constrained by the underlying image-to-flow generator. Walker et al. (2016) also consider image-to-motion generation but take a different approach: they use DCT-based linear compression to encode trajectory offsets and employ a VAE to generate future point trajectories directly from static images. This approach has two key limitations. First, the linear DCT compression lacks both the expressivity and the regularized latent structure that would facilitate effective generative modeling. Second, their evaluation metrics do not assess whether the sampled future trajectories are physically plausible or whether the distribution of generated samples accurately captures the true multi-modal nature of possible outcomes. Li et al. (2018) also consider generating motion from still image, but focuses on image-to-video generation rather than motion prediction itself. They formulate image-to-video task as a two-phase process: first predicting future optical flow maps from a static image, then translating these flow maps into RGB frames. This approach has several key limitations. First, optical flow as a motion representation cannot guarantee long-term temporal consistency due to independent frame-pair estimation, leading to error accumulation, and fails under occlusion. In contrast, we use point trajectories, which state-of-the-art trackers like CoTracker (Karaev et al., 2024a) and AllTracker Harley et al. (2025) estimate jointly to ensure better temporal consistency and tracking through occlusions. Second, while both methods employ VAEs, their roles differ fundamentally: they use a VAE to directly generate flow maps, whereas we use a VAE to construct a regularized latent space for generative modeling with rectified flow, providing better structure for sampling diverse futures. Third, similarly to Walker et al. (2016), their evaluation relies solely on RMSE between predicted and ground-truth flow maps, whereas we rigorously assess whether sampled futures preserve physical plausibility (rigidity), capture the true distribution of possible outcomes (FVMD), and generalize to out-of-distribution object shapes.

**Measuring generation quality.**   Several metrics have been proposed to evaluate the quality of image and video generation. These include IC (Salimans et al., 2016), FVD (Unterthiner et al., 2018), VBench (Huang et al., 2024), and VideoPhy (Bansal et al., 2024). Each set of metrics captures different aspects of generation quality, such as fidelity, diversity, and physical plausibility. VBench (Huang et al., 2024), in particular, considers the quality of motion by assessing whether generated frames can be adequately interpolated with a pre-trained video interpolation model. This, however, only checks whether the motion is smooth and predictable from adjacent frames. We concentrate on motions that are both accurate and plausible.

## 3 METHOD

In our work, a point trajectory is a sequence of 2D pixel coordinates $((x_0, y_0), (x_1, y_1), \ldots, (x_T, y_T))$ describing a point's position over time, starting from its initial position $(x_0, y_0)$. We formulate image-based motion forecasting as the problem of predicting the trajectories of a quasi-dense grid of image pixels, representing the motion of the objects contained in a given image.

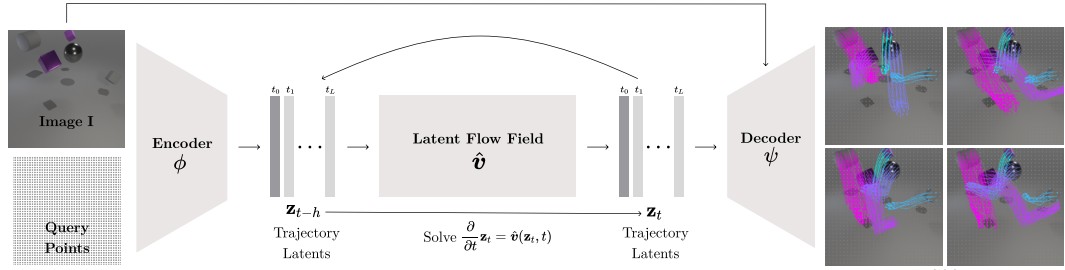

Figure 2: **Method overview**. We generate future trajectories from a single image using a flow matching denoiser that operates in the latent space of a trajectory VAE.

Formally, the image is a tensor $\mathbf{I} \in \mathbb{R}^{H \times W \times C}$, where $C = 3$ is the number of color channels, and $H$ and $W$ are the image height and width in pixels. We predict the motion of the image points for $T$ steps. The density of the tracks is controlled by sampling a grid of tracked points with stride $s \geq 1$. Hence, the trajectories form a tensor $\mathbf{x} \in \mathbb{R}^{\frac{H}{s} \times \frac{W}{s} \times T \times 2}$. Our goal is to predict $\mathbf{x}$ from $\mathbf{I}$.

Because this prediction problem is highly ambiguous, we cast it as learning a *conditional distribution* over possible trajectories. Thus, we take $\mathbf{x}$ to be a sample from a random variable $\mathbf{X}$ and learn the distribution $p(\mathbf{X} \mid \mathbf{I})$.

This task is similar to video generation, except that, instead of generating RGB values, we generate point coordinates. Consequently, we construct this generator using techniques similar to those underlying modern video generators. In particular, we adopt a latent flow matching approach to trajectory prediction (Fig. 2). This involves encoding the trajectories in a compact latent space (Section 3.1) and then learning a denoising neural network operating in this space (Section 3.2). Both components use a similar neural network architecture (Section C.1).

## 3.1 TRAJECTORY LATENT SPACE

Rather than generating the trajectories $\mathbf{x}$ directly, we generate a corresponding latent code $\mathbf{z}$, obtained using a variational autoencoder (VAE) (Kingma & Welling, 2013). The VAE comprises an encoder function $\phi$ mapping $\mathbf{x}$ to (the mean and variance of) a latent code $\mathbf{z}$ and a decoder function $\psi$ mapping $\mathbf{z}$ back to $\mathbf{x}$.

The code $\mathbf{z} \in \mathbb{R}^{\frac{H}{rs} \times \frac{W}{rs} \times T \times D}$ is a tensor with a shape similar to $\mathbf{x}$ but with an additional spatial downsampling factor $r \in \mathbb{N}$ and a latent dimension $D$. Since our generative model operates on short windows ($T \in \{16, 24, 30\}$), we do not compress time.

Because the trajectory grid is only (quasi-)dense and may not cover all parts of an object, we provide the corresponding image $\mathbf{I}$ as input to both the encoder and decoder. This auxiliary input improves the model's ability to reason about object boundaries, shapes, and geometry. The encoder $(\mu_{\mathbf{Z}}, \sigma_{\mathbf{Z}}) = \phi(\mathbf{x} \mid \mathbf{I})$ is thus a mapping $\phi : \mathbb{R}^{\frac{H}{s} \times \frac{W}{s} \times T \times 2} \times \mathbb{R}^{H \times W \times C} \to \mathbb{R}^{\frac{H}{rs} \times \frac{W}{rs} \times T \times 2} \times \mathbb{R}^{\frac{H}{rs} \times \frac{W}{rs} \times T \times 2}$, outputting the parameters of a Gaussian distribution $\mathcal{N}_{\phi(\mathbf{x}|\mathbf{I})}$ with mean $\mu_{\mathbf{Z}}$ and variance $\sigma_{\mathbf{Z}}$. The decoder $\mathbf{x} = \psi(\mathbf{z} \mid \mathbf{I})$ is a mapping $\psi : \mathbb{R}^{\frac{H}{rs} \times \frac{W}{rs} \times T \times D} \times \mathbb{R}^{H \times W \times C} \to \mathbb{R}^{\frac{H}{s} \times \frac{W}{s} \times T \times 2}$, outputting the mean of the reconstructed trajectories.

The model is trained with a $\beta$-VAE objective (Higgins et al., 2017), using a Huber loss $L_\delta$ for reconstruction and a KL divergence to regularize the Gaussian code posterior $\mathcal{N}_{\phi(\mathbf{x}|\mathbf{I})}$ with respect to the normal distribution $\mathcal{N}_0$. For a training sample $(\mathbf{x}, \mathbf{I})$, the loss is:

$$\mathcal{L}_{\beta\text{-VAE}}(\phi, \psi, \mathbf{x}, \mathbf{I}) = \mathbb{E}_{\mathbf{z} \sim \mathcal{N}_{\phi(\mathbf{x}|\mathbf{I})}} \left[ L_\delta \left( \mathbf{x}, \psi(\mathbf{z} \mid \mathbf{I}) \right) \right] + \beta \cdot \mathbb{D}_{\text{KL}} \left( \mathcal{N}_{\phi(\mathbf{x}|\mathbf{I})} \mid \mathcal{N}_0 \right).$$

We implement both the encoder and decoder using a symmetric spatio-temporal transformer, discussed in Section C.1. To downsample, we fold the temporal dimension into the batch dimension, encode (spatial) trajectories, and patchify. To encode trajectories, we concatenate normalized spatial coordinates $(x, y)$ with their learnable Fourier features (Li et al., 2021). These encoded trajectories are then patchified with a non-overlapping 2D convolution with kernel size $r$ and stride $r$, which

outputs the number of channels matching our model embedding dimension. Next, we unfold the temporal dimension to obtain a tensor $\mathbf{h}_{\text{enc}} \in \mathbb{R}^{T \times \frac{H}{rs} \times \frac{W}{rs} \times D}$, where $D$ is the model dimension. A spatio-temporal transformer encoder processes $\mathbf{h}_{\text{enc}}$ and finally linearly projects the embeddings to the mean and variance of the latent code distribution, yielding a latent representation $\mathbf{z} \in \mathbb{R}^{\frac{H}{rs} \times \frac{W}{rs} \times T \times D}$ after sampling with the reparameterization trick (Kingma & Welling, 2013). To decode, we linearly project the latent code to a hidden representation $\mathbf{h}_{\text{dec}} \in \mathbb{R}^{T \times \frac{H}{rs} \times \frac{W}{rs} \times D}$. This representation is then processed with a spatio-temporal transformer decoder, matching the encoder. Finally, the outputs of the decoder are projected to the trajectory patch dimension and assembled into the trajectory grid $\mathbf{x} \in \mathbb{R}^{\frac{H}{s} \times \frac{W}{s} \times T \times 2}$.

## 3.2 SAMPLING TRAJECTORIES USING FLOW MATCHING

Having mapped the trajectories to a latent space, we can now learn a generative model for the latent code $\mathbf{z}$, namely the conditional distribution $p(\mathbf{Z}|\mathbf{I})$. We do so with *rectified flow / flow matching* formulation (Lipman et al., 2022; Liu et al., 2023b).

Briefly, let $\mathbf{Z}_1 = \mathbf{Z}$ be distributed as $p(\mathbf{Z}|\mathbf{I})$, and let $\mathbf{Z}_0 \sim \mathcal{N}(0, I)$ be normally distributed. Define a straight path $\mathbf{z}_t = (1-t)\mathbf{z}_0 + t\mathbf{z}_1$ connecting the noise sample $\mathbf{z}_0$ to the target sample $\mathbf{z}_1$. The velocity of the path at any intermediate point $\mathbf{z}_t$ is constant and given by $\boldsymbol{v}(\mathbf{z}_t, \mathbf{z}_0, t) = \frac{\partial}{\partial t}\mathbf{z}_t = \mathbf{z}_1 - \mathbf{z}_0$. We learn a neural network $\hat{\boldsymbol{v}}(\mathbf{z}_t, \mathbf{I}, t)$ that estimates the expected velocity with respect to all paths passing through $\mathbf{z}_t$ at time $t$ (conditioned on $\mathbf{I}$), by minimizing the Rectified Flow (RF) loss:

$$\mathcal{L}_{\text{RF}}(\hat{\boldsymbol{v}}) = \mathbb{E}_{\mathbf{z}_0, (\mathbf{z}_1, \mathbf{I}), t} \left[ \|\hat{\boldsymbol{v}}(\mathbf{z}_t, \mathbf{I}, t) - \boldsymbol{v}(\mathbf{z}_t, \mathbf{z}_0, t)\|_2^2 \right], \quad \mathbf{z}_t = (1-t)\mathbf{z}_0 + t\mathbf{z}_1.$$

Here, $\mathbf{z}_0 \sim \mathcal{N}(0, I)$ is a normal sample, $(\mathbf{z}_1, \mathbf{I})$ is a training sample, and $t \sim \text{Uniform}[0, 1]$ is a random time step. At test time, we draw samples from the target distribution $p(\mathbf{z}|\mathbf{I})$ by first sampling $\mathbf{z}_0 \sim \mathcal{N}(0, I)$ from the normal distribution and then moving it towards $\mathbf{z} = \mathbf{z}_1$ along the path defined by the velocity field $\hat{\boldsymbol{v}}(\mathbf{z}_t, \mathbf{I}, t)$, which amounts to integrating an ODE.

## 3.3 TRAINING

During training, only *one ground truth future* is observed *for each initial condition*, which reflects the setting where real data provides only one ground truth future. At *inference time*, however, we aim to produce *multiple plausible hypotheses*. This is challenging because the model must infer the existence of multiple possible futures from nearby training examples. However, simple interpolation between training samples is not necessarily physically plausible. For example, naive interpolation between rigid motions does not, in general, yield a rigid motion. This setup is very different from training text-to-image/video models, where we have thousands of examples matching a caption.

## 3.4 MEASURING THE GENERATION QUALITY

Predicting possible scene motion from a single image is a highly ambiguous task. Hence, regression metrics, which assume a single possible ground truth output, are not suitable. We instead report the *Best-of-$K$* Mean Square Error (MSE), which is the lowest error obtained between pairs of generated and simulated trajectories $\mathbf{x}$ for each image $\mathbf{I}$. To compute this, we simulate $K$ possible trajectories for each image $\mathbf{I}$ (randomizing the initial velocities) and compare them to $K$ generated trajectories $\mathbf{x}$.

We also assess the *statistical plausibility* of the generated trajectories using *motion distributional metrics*. In particular, we use the *FVMD* (Liu et al., 2024) metric, which calculates the Fréchet distance between generated and simulated trajectories using histogram-based features. However, because FVMD compares the generated and simulated versions of the *marginal* distribution $p(\mathbf{X}) = \mathbb{E}_{\mathbf{I}}[p(\mathbf{X}|\mathbf{I})]$, it does not evaluate whether the generated motions $\mathbf{X}$ are plausible for a *specific* image $\mathbf{I}$. To address this, we also compute the FVMD image-wise (*FVMD (S)*) to evaluate the conditional distribution $p(\mathbf{X}|\mathbf{I})$, which is feasible since we generate and simulate multiple trajectories per image.

Finally, we evaluate the *physical plausibility* of the generated motion. Using the mask of each object (available as part of the simulated data), we identify which trajectories belong to each object. We then measure whether these 2D trajectories could have arisen from an underlying rigid 3D object. As a 'rigidity' metric, we repurpose the method of (Karazija et al., 2024), which posits that trajectories stacked into a matrix should exhibit low-rank structure. We define the *LRTL* score as the mean

Frobenius norm between predicted trajectories, collected into a matrix, and their truncated SVD reconstruction at rank 5. Intuitively, if objects fail to maintain shape or if not all points move cohesively, the LRTL score increases because the reconstruction cannot adequately represent such linearly independent motions.

FVMD and LRTL are complementary metrics. For instance, FVMD may fail to detect if point velocities are shuffled within the spatio-temporal window used to compute motion statistics. On the other hand, LRTL is minimized if there is no motion at all, even if the generated trajectories are dissimilar to the ground truth.

## 4 EXPERIMENTS

In this section, we begin by comparing our method with regression-based baselines to highlight the effectiveness of combining stochastic trajectory generation with grid-based global scene reasoning. We then present comparisons with generative methods. We show that our method surpasses a generative trajectory baseline, underscoring the advantages of our proposed architecture. We also present results comparing with large-scale image-to-video generators, which we apply for the motion prediction task. We further study why RGB video is a suboptimal proxy for modeling motion, underscoring the importance of point trajectories as the appropriate modality for motion generation. Finally, we conclude with results on real-world data.

### 4.1 COMPARISON WITH REGRESSION METHODS

We compare our method with ATM (Wen et al., 2024a) and Tra-MoE (Yang et al., 2025), regression-based trajectory predictors, using the LIBERO robotics datasets. For these methods, we use their official implementation and the checkpoints. ATM and Tra-MoE reported the success rate of policies trained on generated trajectories, without directly assessing the quality of the generated trajectories themselves. Here, we are interested in the *motion* prediction problem; we thus adopt the MSE evaluation metric, since we have only one ground truth. Since ATM and Tra-MoE regress trajectories from the initial frame and a text instruction, we extend our method with text conditioning, using the same BERT model to process text as the baselines. Details are in Section D.4.1. For evaluation, we favour the baselines by choosing trajectories according to the number of points and filtering scheme they employ during training. In contrast, our method directly predicts trajectories at every other pixel, which include evaluation points as a subset. To fairly compare with regression baselines, we compute results for our method in three ways:

MeanT   Calculate the average of $k$ samples to form the average predicted trajectory. The mean prediction is used to evaluate metrics, checking whether a method recovers the correct mode.

Mean   Compute metric for each $k$ samples, averaging the results.

Min   Compute metric for each $k$ samples, taking the minimum of the results.

Table 1: **Comparison with ATM** on LIBERO datasets using MSE.

| | LIBERO-90 | | LIBERO-10 | |
|---|---|---|---|---|
| **Model** | **Side** | **Effector** | **Side** | **Effector** |
| ATM ($k = 1$) | 23.07 | 67.37 | 31.02 | 69.96 |
| **Ours (MeanT, $k = 8$)** | **16.70** | **52.70** | **23.69** | **58.35** |
| **Ours (Mean, $k = 8$)** | 18.32 | 60.47 | 26.71 | 66.35 |
| **Ours (Min, $k = 8$)** | 10.99 | 32.01 | 13.86 | 35.93 |

Tables 1 and 2 show that our method considerably outperforms the baselines, whether we form an average trajectory (MeanT) or consider individual samples for expected performance (Mean) or the best-case scenario (Min). This suggests that modeling uncertainty in motion generation is more important than domain-specific architectural changes, such as the mixture of experts in Tra-MoE. Results for different $k$ are in Section A.1. We also study qualitative outputs (Fig. 3), where our method can sample diverse yet consistent predictions. We attribute this to modeling full scene motion using a grid. This is particularly important given the effector view, where camera motion is uncertain.

Table 2: **Comparison with Tra-MoE** on LIBERO datasets using MSE.

| Model | GOAL | | OBJECT | | SPATIAL | | LIBERO-10 | |
|---|---|---|---|---|---|---|---|---|
| | Side | Effector | Side | Effector | Side | Effector | Side | Effector |
| Tra-MoE ($k = 1$) | 27.56 | 105.92 | 14.07 | 48.78 | 37.62 | 88.22 | 40.54 | 82.23 |
| **Ours (MeanT, $k = 8$)** | **15.85** | **71.41** | **8.94** | **30.65** | **15.46** | **54.71** | **26.20** | **63.62** |
| **Ours (Mean, $k = 8$)** | 17.46 | 87.38 | 10.26 | 36.91 | 16.91 | 65.48 | 31.73 | 78.50 |
| **Ours (Min, $k = 8$)** | 10.52 | 37.41 | 5.57 | 18.08 | 10.95 | 33.25 | 13.52 | 34.58 |

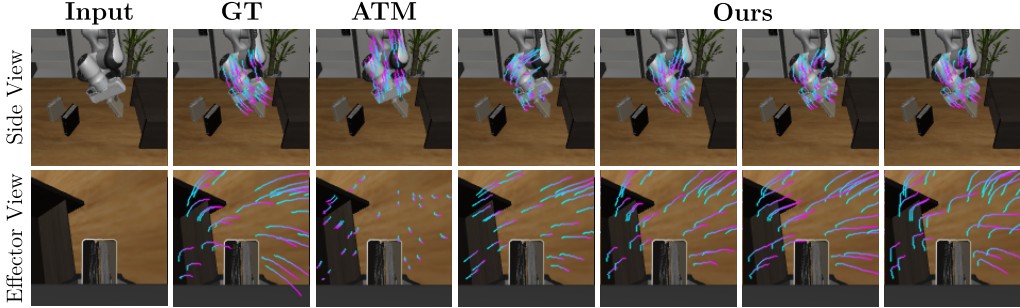

Figure 3: **Qualitative comparison on LIBERO** for the task *"Pick up the book on the right and place it under the cabinet shelf"*. Unlike the baseline (ATM), we sample diverse predictions for the entire scene, particularly beneficial for the uncertain effector view, where camera is attached to the effector.

## 4.2 COMPARISON WITH GENERATIVE METHODS

Table 3: **Motion generation quality on Kubric** (Greff et al., 2022). Our method shows better adherence to the ground truth motion over multiple metrics. [†]- model fine-tuned to Kubric dataset.

| Model | FVMD | FVMD (S) | Best of K | LRTL |
|---|---|---|---|---|
| *Diffusion-Based Trajectory Generators* | | | | |
| Track2Act (Bharadhwaj et al., 2024b) | 16735 | 22509 | 250.8 | 15.8 |
| **Ours (L)** | **13745** | **17838** | **127.0** | **14.1** |
| *Diffusion-Based Video Generators* | | | | |
| WAN 14B (Wan et al., 2025) | 34573 | 42987 | 184.6 | 35.1 |
| Stable Video Diffusion (Blattmann et al., 2023) | 30173 | 39494 | 235.7 | 37.2 |
| LTX-Video (HaCohen et al., 2024a) | 24722 | 32019 | 205.1 | 17.0 |
| WAN 1.3B (Wan et al., 2025) | 23608 | 30712 | 192.6 | 42.1 |
| DynamicCrafter[†] (Xing et al., 2023) | 41398 | 50123 | 239.9 | 51.8 |
| Stable Video Diffusion[†] (Blattmann et al., 2023) | 17099 | 22799 | 152.2 | 30.1 |
| WAN 1.3B[†] (Wan et al., 2025) | 14864 | 20010 | 162.8 | 26.6 |

Table 4: **Motion generation quality on an out-of-distribution version of Kubric**. [†]- model fine-tuned to Kubric dataset.

| Model | FVMD | FVMD (S) | Best of K | LRTL |
|---|---|---|---|---|
| *Diffusion-Based Trajectory Generators* | | | | |
| Track2Act (Bharadhwaj et al., 2024b) | 15751 | 19608 | 278.6 | 19.7 |
| **Ours (L)** | **12221** | **14949** | **127.2** | **15.9** |
| *Diffusion-Based Video Generators* | | | | |
| DynamicCrafter[†] (Xing et al., 2023) | 43248 | 49092 | 230.5 | 58.8 |
| Stable Video Diffusion[†] (Blattmann et al., 2023) | 16113 | 19780 | 127.7 | 31.7 |
| WAN 1.3B[†] (Wan et al., 2025) | 13253 | 16547 | 128.2 | 27.3 |

Table 5: **User study**. Our model is ranked better than (SVD and WAN 1.3B) 52% of the time.

| Model | 1st (%) | 2nd (%) | 3rd (%) | ELO |
|---|---|---|---|---|
| SVD[†] | 20 | 38 | 42 | 929 |
| WAN 1.3B[†] | 28 | 36 | 36 | 987 |
| **Ours** | 52 | 26 | 22 | 1084 |

We now conduct an in-depth evaluation of our proposal with various generative approaches that model trajectories or pixels for predicting motion. We train our model on the MOVi-A variant of Kubric (Greff et al., 2022), which features geometric primitives of various colours being launched into the centre of the scene, falling, and colliding. For evaluation, we generate a new split of data containing 16 scenes. In each scene, we sample 64 different initial velocity configurations, resulting in 1,024 unique evaluation settings. We condition both our method and the baselines to predict the next 23 frames based on an initial frame. CoTracker3 (Karaev et al., 2024a) is used to obtain trajectories following video generation for baselines. Tables 3 and 4 contain the results.

We compare against Track2Act, a method that also generates point trajectories. Specifically, Track2Act represents an image as a single vector encoded by a small ResNet18, flattens trajectories, and performs standard attention. In contrast, we perform spatio-temporal cross-attention with all patch tokens from the conditioning frame, encoded by DINOv2 (Oquab et al., 2024). Our method significantly outperforms Track2Act, despite it being more than twice as large. We attribute this to the stronger inductive bias offered by cross-attention. This provides more information about object location and geometry, resulting in lower LRTL. Further, in Table 14, we show that the same conclusions hold when we train our method on pseudo ground truth trajectories from CoTracker3.

Next, we explore whether image-to-video generators can solve our task without fine-tuning. Despite significant pretraining and assumed generality, they struggle. We observe that LTX (HaCohen et al., 2024a) shows a low LRTL score but high distributional metrics. Upon inspection, the model struggles to maintain the object shape and generates sudden motions, causing point tracking to fail.

We further fine-tune all three video baselines on the Kubric dataset to minimize domain gaps and give the generators an opportunity to learn the motion patterns and invariants present in the data. Even after this adjustment, our model continues to outperform all the baselines by a clear margin. Notably, trajectory-based methods tend to produce far more rigid motion, as reflected by their substantially lower LRTL. This result supports our hypothesis that RGB-space generation introduces excessive overhead, leading to implausible non-rigid motion, visible as object shape inconsistencies (Figure 1).

We also carry out a similar quantitative evaluation using the out-of-distribution version of the Kubric dataset, which we generate using a different set of object primitives. Table 4 shows that our method performs favourably in this setting as well, though all methods show increased metrics, indicating slightly affected performance out-of-distribution.

Finally, we validate the results of our quantitative evaluation with a user study. We choose the methods according to Best-Of-K. We show 16 different scenes to 20 users and ask them to rank three models in order of preference for what they think is the most plausible, realistic depiction of future scene motion. More details are in the supplementary. We report the results in Table 5. We find that our model ranks as the best 52% of the time, with an ELO score significantly higher than other methods.

### 4.3 SIGNIFICANCE OF MODALITY FOR MOTION GENERATION

Motivated by the results in Tables 3 and 4 and the qualitative evidence in Fig. 1, we hypothesize that motion implausibility arises from the overhead associated with pixel-level RGB generation. Specifically, RGB synthesis requires the model to allocate its capacity to low-level appearance factors such as lighting and texture, thereby reducing its focus on motion accuracy or physical plausibility. To evaluate this hypothesis, we fix our base architecture and ablate only the output modality. We downsample RGB such that the latent shapes of the RGB generator and the trajectory generator are comparable. We first verify that CoTracker3 reliably extracts motion from video generators (Table 23) and that RGB VAEs achieve high reconstruction accuracy (Table 7). This confirms that motion errors

Table 6: **Switching modality** from RGB to point trajectories considerably improves motion generation quality. Joint diffusion of RGB and trajectories substantially improves the motion quality extracted from generated RGB.

| Latent Space | Latent Shape | FVMD | FVMD (S) | Best-Of-K | LRTL |
|---|---|---|---|---|---|
| *Kubric (In-Distribution)* | | | | | |
| SVD | 24×16×16×4 | 20589 | 26789 | 195 | 48.5 |
| SD3.5 | 24×16×16×16 | 16592 | 21869 | 147 | 33.7 |
| WAN | 7×16×16×16 | 17320 | 22867 | 160 | 31.1 |
| SD3.5 + Tracks | 24×24×16×16 | 15399 | 20414 | 136 | 28.2 |
| **Tracks** | 24×16×16×8 | **12221** | **14950** | **127** | **15.9** |
| *Kubric (Out-Of-Distribution)* | | | | | |
| SVD | 24×16×16×4 | 18740 | 22865 | 155 | 49.3 |
| SD3.5 | 24×16×16×16 | 13661 | 17004 | 115 | 28.0 |
| WAN | 7×16×16×16 | 15155 | 18761 | 132 | 30.0 |
| SD3.5 + Tracks | 24×24×16×16 | 13386 | 16694 | **109** | 24.6 |
| **Tracks** | 24×16×16×8 | **12062** | **14748** | 129 | **18.4** |

Table 7: **RGB VAEs reconstruct** Kubric with minimal errors.

| VAE | PSNR | SSIM | LPIPS |
|---|---|---|---|
| *Kubric (In-Distribution)* | | | |
| SVD | 36.06 | 0.97 | 0.04 |
| SD3.5 | 37.31 | 0.99 | 0.02 |
| WAN | 37.36 | 0.97 | 0.03 |
| *Kubric (Out-Of-Distribution)* | | | |
| SVD | 31.76 | 0.94 | 0.05 |
| SD3.5 | 33.86 | 0.97 | 0.03 |
| WAN | 32.88 | 0.95 | 0.04 |

in generated videos stem from unrealistic motion, not tracking or autoencoding artifacts. We train all RGB and trajectory generators under the same setup: identical training procedure, duration, and hardware. Table 6 strongly supports our hypothesis. In particular, trajectory-based flow matching generates motion that is significantly closer to the ground truth distribution while better respecting the rigidity invariant. Since our trajectory model operates on latents of comparable dimensionality (Table 6), this is not due to the reduced dimensionality, but rather to the superior choice of modality.

Motivated by this observation, we then explore image-to-video generation as a downstream application and present preliminary evidence that our trajectory generation method can be used to enhance motion quality in synthesized videos. To this end, as before, we adopt a fixed base denoising architecture while varying the input modality, For video generation, we select the StableDiffusion3.5 (SD3.5) RGB latent space, which demonstrates superior motion quality compared to SVD and WAN (Table 6). Then, we train a specialized VAE that encodes 32×32 trajectory grids conditioned on 128×128 images. This trajectory VAE is the same as our 256×256 VAE but operates at lower resolution to align the latent tensor dimensions with the RGB VAE.

We then train two generative models: (1) an RGB-only model that generates video from the SD3.5 latent space (SD3.5 in Table 6), and (2) a joint model that simultaneously generates video and trajectories from both the SD3.5 RGB latent space and the trajectory VAE latent space (SD3.5 + Tracks in Table 6). The RGB-only model generates 16-channel SD3.5 latents, while the joint model generates 24-channel inputs latents by concatenating RGB latents (16 channels) with trajectory latents (8 channels). This concatenation is sensible because the trajectory VAE grid structure matches the RGB VAE grid structure. In the joint model, we discard the directly generated trajectories and extract trajectories from generated RGB using CoTracker3. This evaluation measures the motion quality of the generated RGB frames themselves, rather than the explicitly generated trajectories.

As shown in Table 6, jointly diffusing RGB frames and trajectories significantly improves the quality of motion extracted from the generated RGB. This improvement is consistent across all metrics. The substantial gain in the rigidity proxy metric (LRTL) indicates that the generated motion is more rigid and thus more physically plausible. This result further confirms the advantage of directly generating motion represented as point trajectories.

## 4.4 RESULTS ON REAL-WORLD DATA

We further evaluate our method on two real-world datasets: Physics101 and Cityscapes. Physics101 enables study of object physical properties from unlabeled videos, while Cityscapes provides urban driving scenarios captured across 50 cities with diverse egocentric motion and viewpoints. Together, these datasets allow us to examine a broader range of physical phenomena and interactions while assessing our method's performance on motion forecasting under unconstrained, real-world conditions.

**Physics101:** Physics101 consists of roughly 10000 video clips containing 101 objects of various materials and appearances (shapes, colors, and sizes). We evaluate five different physical scenarios, namely fall, liquid, multi, ramp, and spring. Our evaluation set contains 1450 different initial conditions, with a single ground truth per initial condition. Due to the high cost of sampling from video generators (Table 22), we sample once from each method and compare with the single pseudo ground truth from CoTracker3. As only a single ground truth is available, we report MSE.

Results in Table 8 show that overall our method shows comparable or better performance than large-scale fine-tuned WAN, with better performance overall. Analysis in Figure 6 shows that our method produces fewer outliers. In many cases, it achieves 10× lower MSE.

Table 8: **Comparison with WAN** on **Physics101** using MSE. Our method outperforms WAN overall and in $3/5$ evaluated scenarios, including the most complex *Multi* scenario.

| Model | Physical Scenario | | | | | |
| | Fall | Liquid | Multi | Ramp | Spring | Overall |
|---|---|---|---|---|---|---|
| WAN (1.3B) | **16.05** | **4.48** | 21.88 | 37.53 | 70.48 | 30.08 |
| **Ours (B)** | 19.78 | 6.00 | **15.65** | **36.35** | **65.31** | **28.62** |

**Cityscapes:** Our Cityscapes training set consists of approximately 3000 video clips of 30 frames of resolution $224 \times 448$, captured across 19 different cities, towns, and open roads. For evaluation, we use 500 validation clips from three unseen locations. We employ CoTracker3 to generate pseudo ground truths for both training and evaluation. We evaluate on $28 \times 56$ trajectory grids, sampling from each method 8 times, every time with a different seed. Given that only a single pseudo ground truth future is available per clip, we adopt the evaluation protocol from Section 4.1.

As shown in Table 9, our method substantially outperforms both Track2Act and fine-tuned WAN across all metrics, producing more accurate motion overall. Interestingly, WAN frequently hallucinates in scenes where the car makes a turn (Figure 11), generating RGB outputs that confuse the point tracker, while Track2Act often fails to account for (distant) objects in the scene, such as pedestrians or cars. These limitations may be a consequence of their limited input image conditioning: Track2Act uses a small pre-trained ResNet to represent the entire scene as a single vector for adaptive normalization. In contrast, we condition on patch-level tokens from DINO through cross-attention in every block, providing richer information about the input image. Figure 12 demonstrates that our method also works on soft bodies, such as pedestrians.

Table 9: **Comparison on Cityscapes** using MSE. Our method significantly outperforms both Track2Act and the fine-tuned WAN 1.3B in motion forecasting.

| Model | MeanT | Mean | Min |
|---|---|---|---|
| Track2Act | 7037.88 | 9305.63 | 4393.58 |
| WAN (1.3B) | 2650.59 | 3495.43 | 1799.04 |
| **Ours (L)** | **1475.68** | **1565.06** | **1240.6** |

Finally, in Section A.4, we present a qualitative study to showcase our method's ability to generalize from synthetic training data from Physion (Bear et al., 2022) to real-world scenes that we recorded. Reproducibility details, evaluation procedures, and design choice studies are in the Appendix.

## 5 CONCLUSION

In this work, we address motion anticipation from a single image by formulating it as the conditional generation of dense trajectory grids. Our results highlight the benefits of modeling uncertainty in the motion of the entire scene over prior trajectory regressors and generators. We extensively evaluate our approach in simulated settings, assessing diversity, physical consistency, and user preference. We also show that large-scale pretrained video generators underperform in motion prediction, even in simple physical scenarios such as falling blocks or mechanical interactions, on simulated or real data. By switching the output modality of our method, we experimentally show that this limitation arises from the overhead of generating RGB pixels rather than directly modeling motion trajectories.

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

APPENDIX

In this supplementary material, we provide the following:

1. Ethics Statement
2. Additional results and comparisons
3. Interactive animations (provided in the `visualizations` folder):

   (a) Qualitative comparison versus the video-generation baselines on Kubric.
   (b) Qualitative comparison versus the robotics baselines.
   (c) More real-world examples.

4. Detailed implementation information, including model architecture, training and sampling hyperparameters, and reproducibility settings such as required hardware and estimated reproduction time.
5. Details about the user study.
6. User study scenes (provided in the `user_study_scenes` folder):
7. The Use of Large Language Models (LLMs) Statement
8. Limitations

# A  ADDITIONAL RESULTS AND COMPARISONS

## A.1  MORE QUANTITATIVE RESULTS ON LIBERO

In Tables 10 and 11, we report the performance of our method with different numbers of samples per initial condition ($k$). The results indicate that our approach is fairly robust to the choice of $k$, though using more samples generally leads to better performance across all evaluation metrics, highlighting the advantages of diversity in generation.

Table 10: **Comparison with ATM** on LIBERO datasets using MSE.

| | LIBERO-90 | | LIBERO-10 | |
| Model | Side | Effector | Side | Effector |
|---|---|---|---|---|
| ATM ($k = 1$) | 23.07 | 67.37 | 31.02 | 69.96 |
| $k = 1$ | | | | |
| **Ours (Mean)** | 17.89 | 57.64 | 26.18 | 63.47 |
| **Ours (Min)** | 17.89 | 57.64 | 26.18 | 63.47 |
| $k = 2$ | | | | |
| **Ours (Mean)** | 18.31 | 62.56 | 27.89 | 71.98 |
| **Ours (Min)** | 14.90 | 46.86 | 21.88 | 53.41 |
| $k = 4$ | | | | |
| **Ours (Mean)** | 18.32 | 60.13 | 26.22 | 66.17 |
| **Ours (Min)** | 12.77 | 38.68 | 16.93 | 44.33 |
| $k = 8$ | | | | |
| **Ours (MeanT)** | 16.70 | 52.70 | 23.69 | 58.35 |
| **Ours (Mean)** | 18.32 | 60.47 | 26.71 | 66.35 |
| **Ours (Min)** | 10.99 | 32.01 | 13.86 | 35.93 |

Table 11: **Comparison with Tra-MoE** on LIBERO datasets using MSE.

| Model | GOAL Side | GOAL Effector | OBJECT Side | OBJECT Effector | SPATIAL Side | SPATIAL Effector | LIBERO-10 Side | LIBERO-10 Effector |
|---|---|---|---|---|---|---|---|---|
| Tra-MoE ($k = 1$) | 27.56 | 105.92 | 14.07 | 48.78 | 37.62 | 88.22 | 40.54 | 82.23 |
| $k = 1$ | | | | | | | | |
| **Ours (Mean)** | 16.69 | 91.89 | 10.52 | 34.00 | 17.39 | 61.36 | 34.78 | 72.00 |
| **Ours (Min)** | 16.69 | 91.89 | 10.52 | 34.00 | 17.39 | 61.36 | 34.78 | 72.00 |
| $k = 2$ | | | | | | | | |
| **Ours (Mean)** | 16.77 | 92.18 | 10.10 | 37.40 | 17.08 | 62.92 | 35.86 | 84.51 |
| **Ours (Min)** | 14.03 | 61.68 | 7.88 | 27.56 | 14.07 | 45.11 | 27.06 | 56.88 |
| $k = 4$ | | | | | | | | |
| **Ours (Mean)** | 17.04 | 88.14 | 10.26 | 37.19 | 17.11 | 62.71 | 33.44 | 77.95 |
| **Ours (Min)** | 11.63 | 46.69 | 6.76 | 22.85 | 12.39 | 38.69 | 18.38 | 43.54 |
| $k = 8$ | | | | | | | | |
| **Ours (MeanT)** | 15.85 | 71.41 | 8.94 | 30.65 | 15.46 | 54.71 | 26.20 | 63.62 |
| **Ours (Mean)** | 17.46 | 87.38 | 10.26 | 36.91 | 16.91 | 65.48 | 31.73 | 78.50 |
| **Ours (Min)** | 10.52 | 37.41 | 5.57 | 18.08 | 10.95 | 33.25 | 13.52 | 34.58 |

## A.2 Extension to multiple frames

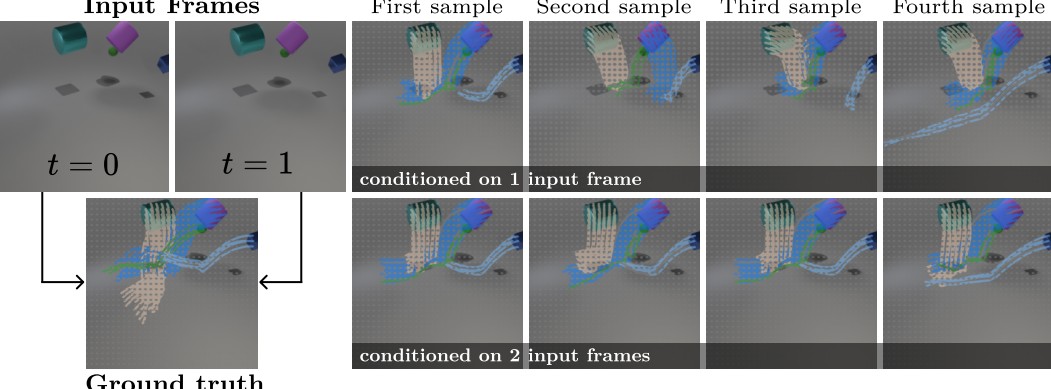

Figure 4: **Trajectory generation diversity with varying number of input frames.** Input frames and the corresponding ground truth future rollout are shown alongside generated samples with different random seeds. Single-frame input produces diverse trajectories reflecting higher uncertainty without velocity information. Two-frame input yields more accurate trajectories with reduced diversity as velocity is observed. Colors distinguish different object instances.

Our model can be easily extended to support video or multi-frame inputs. Currently, we condition on the initial frame using cross-attention over tokens extracted from the initial frame. To incorporate multiple frames, the model would simply cross-attend to tokens from all conditioning frames, requiring no major architectural changes or training setup changes. Please see Section C.1 for details. To demonstrate this, here we train and evaluate a multi-frame version of our method on Kubric.

**Theoretical formulation.** We train a single model that can generate future trajectories from variable number of input images. Formally, the multi-frame model learns conditional distributions of future trajectories given a variable number of input images, $\{p(\mathbf{X} \mid \mathbf{I}_1, \mathbf{I}_2, \ldots, \mathbf{I}_k)\}_{k=1}^{K}$, where $\mathbf{X} \in \mathbb{R}^{T \times \frac{H}{s} \times \frac{W}{s} \times 2}$ are future trajectories on a spatial grid and $K \in \mathbb{N}$ denotes the maximum context length.

**Implementation.** We cross-attend with tokens from all conditioning frames. To encode temporal ordering of the input frames, we add learnable frame index positional encodings to the frame patch tokens before cross-attention. During training, we randomly vary the context length. The trajectory VAE remains unchanged, conditioned only on the initial frame $I_1$.

**Evaluation.** We evaluate the same model with either 1 or 2 input frames on identical test data. The test set is the same as in Section 4.1, including 16 scenes each for in-distribution and out-of-distribution object shapes, with 64 rollouts per scene. Each rollout shares the same initial object placement but varies in initial velocity. Single-frame prediction is highly ambiguous due to unknown initial velocity. With two frames, velocity is observable and the Kubric data generation process is actually deterministic. However, occlusions still introduce some ambiguity in predictions. Therefore, we sample 3 predictions per input configuration (i.e., number of input frames) and report the mean squared error (MSE) between generated and ground truth trajectories. Here, we follow the evaluation protocol from Section A.1 since the Kubric data generation process becomes deterministic when velocity is observed.

**Results.** Table 12 shows that multi-frame samples outperform single-frame samples across all metrics and both dataset settings (i.e., in-distribution or out-of-distribution object shapes). Qualitatively (Figure 4), multi-frame predictions are more accurate and exhibit lower diversity, reflecting reduced uncertainty when velocity is observed. Notably, Figure 5 shows that the model adapts its prediction diversity based on the amount of conditioning information provided. This also demonstrates that generation from single frame is indeed more challenging and that the model knows how to incorporate more information.

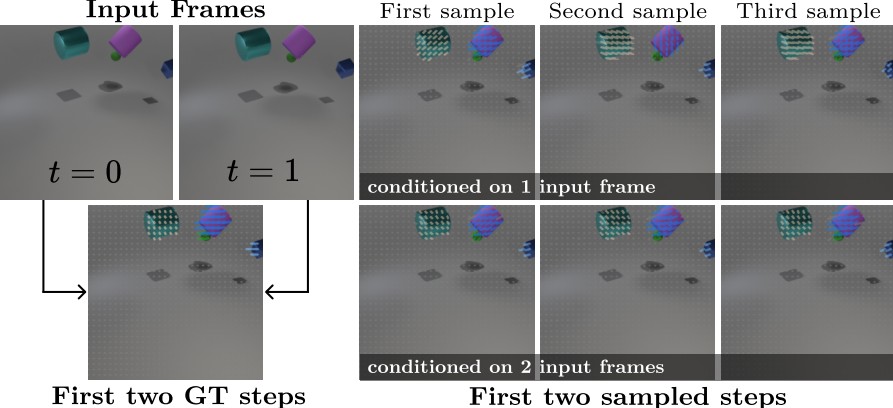

Figure 5: **Initial steps of samples with varying number of input frames.** Input frames and the corresponding ground truth future rollout are shown alongside generated samples with different random seeds. Single-frame input produces diverse trajectories reflecting higher uncertainty without velocity information. Two-frame input enables better estimation of the initial velocity, consequently reducing trajectory diversity and demonstrating that the model correctly incorporates more context.

Table 12: **Trajectory generation performance with varying input frames.** Mean Squared Error (MSE) decreases substantially when using 2 input frames compared to 1 frame across all metrics and datasets. MeanT, Mean, and Min refer to different aggregation methods over predicted trajectories.

|  | MSE (MeanT) | MSE (Mean) | MSE (Min) |
|---|---|---|---|
| *Kubric (In-Distribution)* | | | |
| **1 input frame** | 403.287 | 478.369 | 368.784 |
| **2 input frames** | 253.263 | 264.963 | 226.989 |
| *Kubric (Out-of-Distribution)* | | | |
| **1 input frame** | 275.284 | 305.264 | 255.165 |
| **2 input frames** | 219.097 | 233.268 | 209.901 |

### A.3 MORE QUANTITATIVE RESULTS ON PHYSICS101

Figure 6: **MSE Error Analysis** on Physics101. Compared to WAN, our method does not have extremely wrong predictions (top right corner). Moreover, there are many examples where our method achieves $10\times$ lower MSE (upper left part).

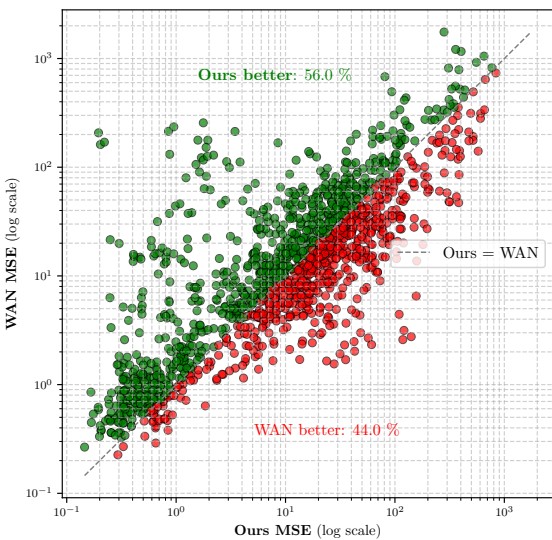

### A.4 QUALITATIVE RESULTS ON PHYSION

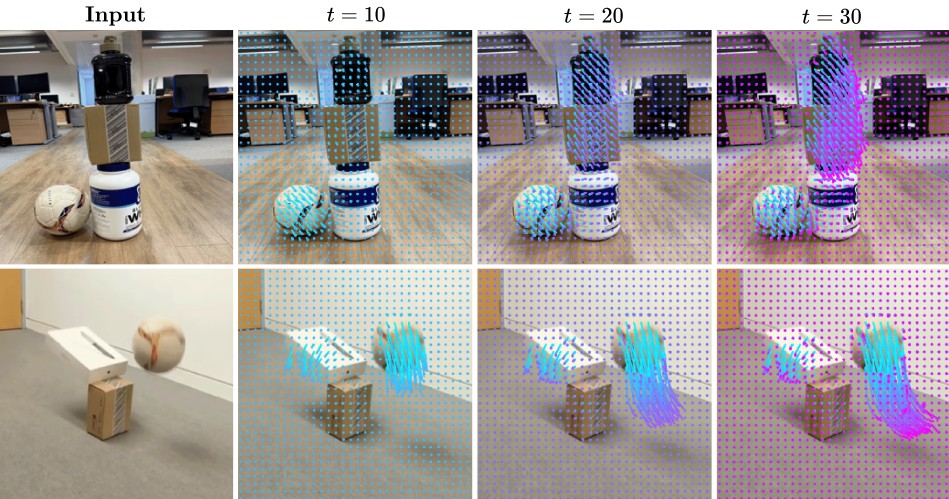

Figure 7: **Real-world Generalization.** Despite training only on synthetic data, without motion blur, our model generalizes to real, unseen objects and different viewpoints in the wild.

Due to limited computational resources, we cannot train on large-scale real-world motion datasets. Instead, we investigate whether our method can be trained on more diverse synthetic data and still be effective in generalizing to real-world scenarios. We train our model on *Physion* Venkatesh et al. (2024), a synthetic dataset, and reserve a set of real-world scenes solely for evaluation. Because our real-world dataset is too small for robust quantitative analysis, we focus on qualitative results, as shown in Fig. 7. More animations are in the supplementary material. In particular, our method

can transfer knowledge of the physical laws learned in the synthetic environment to real scenes with unfamiliar objects, backgrounds, camera viewpoints, and textures. It effectively captures and combines multiple physical phenomena, including gravity, collisions, and force propagation. Although the generalization is not perfect, we believe that this provides strong evidence and a solid foundation for future work in scaling our method.

## A.5 COMPARISON WITH MOTIONMODES

Although MotionModes (Pandey et al., 2024) adresses a different problem (e.g. it requires a segmentation mask for the moving part of the scene), we include it as a baseline because, like our method, it generates predictions conditioned on a single input image.

We use the ground-truth segmentation mask of the entire scene to construct a foreground mask, where pixels corresponding to any object are set to 1 and background pixels to 0. This mask is provided to MotionModes. For fairness, we resize the input image to match their resolution and aspect ration. Then, using their official implementation, we generate the same number of samples as our model, using identical random seeds. Tracks are extracted following the procedure described in their paper and are linearly interpolated from 16 frames (their prediction horizon) to 24 frames (ours). Evaluations are performed on both in-distribution and out-of-distribution samples. Results are in Table 13. Our method clearly outperforms MotionModes.

Table 13: **Comparison with MotionModes on Kubric.** Our method clearly outperforms Motion-Modes.

| Model | FVMD | FVMD (S) | Best of K | LRTL |
|---|---|---|---|---|
| *Kubric (In-Distribution)* | | | | |
| MotionModes(Pandey et al., 2024)24698 | | 30125.0 | 252.7 | 43.7 |
| Ours (B) | 5713.7 | 8309.7 | 146.7 | 8.3 |
| **Ours (L)** | **6470.3** | **9293.0** | **131.4** | **6.4** |
| *Kubric (Out-of-Distribution)* | | | | |
| MotionModes(Pandey et al., 2024)21538 | | 25115.0 | 178.6 | 43.4 |
| Ours (B) | 5118.1 | 6989.3 | 160.4 | 8.5 |
| **Ours (L)** | **5916.9** | **7926.4** | **142.4** | **7.4** |

## A.6 MORE QUALITATIVE RESULTS ON PHYSICS101

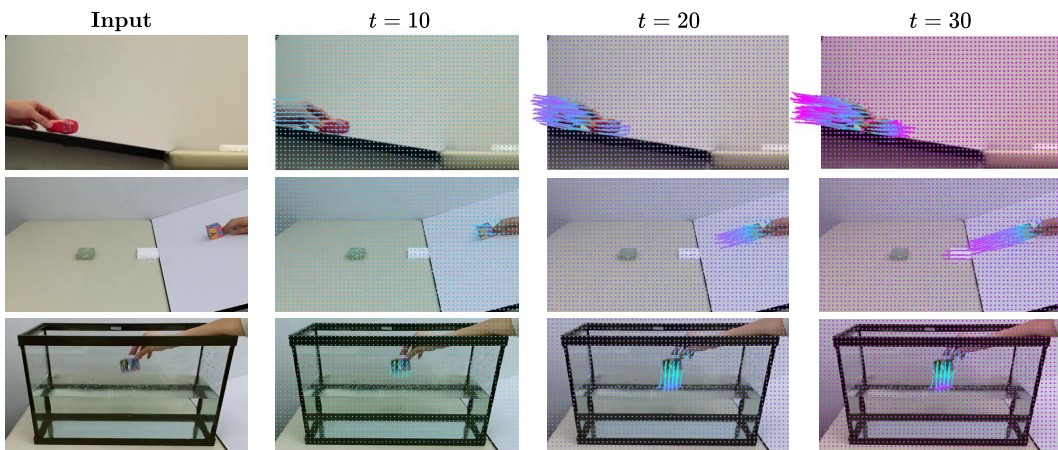

Figure 8: **Qualitative results of our method on Physics101.** Our method can generate the motion of both rigid and non-rigid objects (first row), model force propagation (first and second row), and integrate multiple physical phenomena with different material properties (third row).

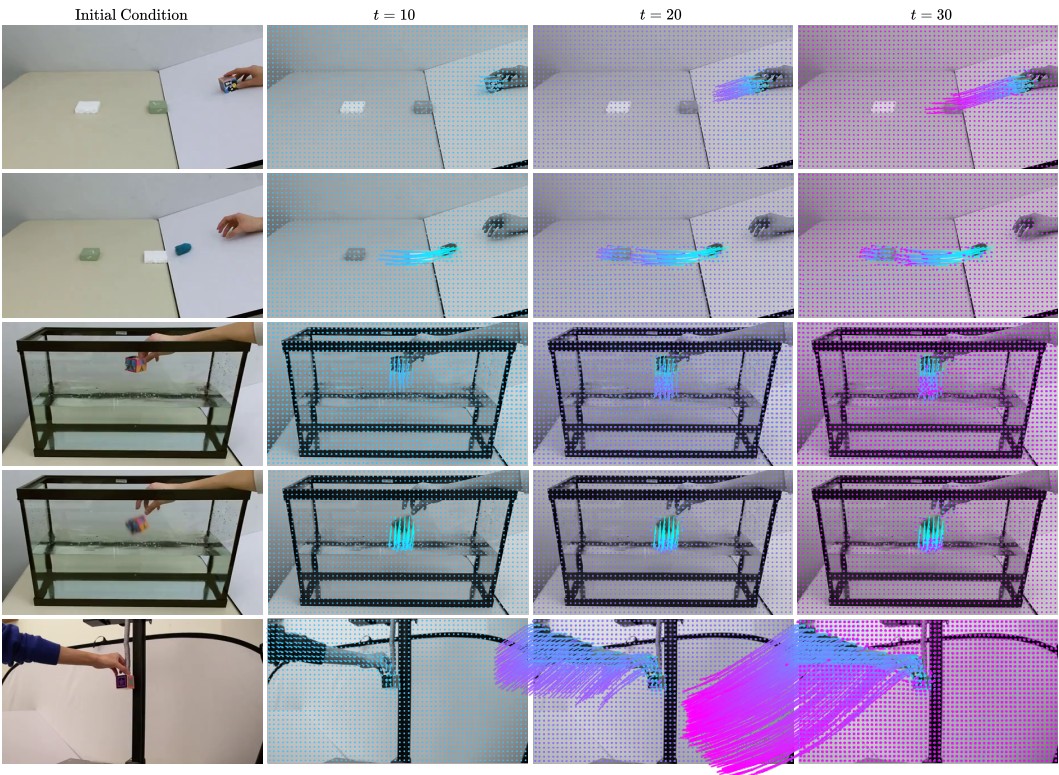

Figure 9: **Qualitative results of our method on Physics101**. The visualization overlays predicted motion trajectories on the initial frame. The color gradient represents temporal progression: blue indicates early timesteps, purple shows intermediate stages, and pink marks later timesteps. Our method can generate the motion of both rigid and non-rigid objects (last three rows), model force propagation (first and second row), and integrate multiple physical phenomena with different material properties (third row and forth row). More examples are on the next page.

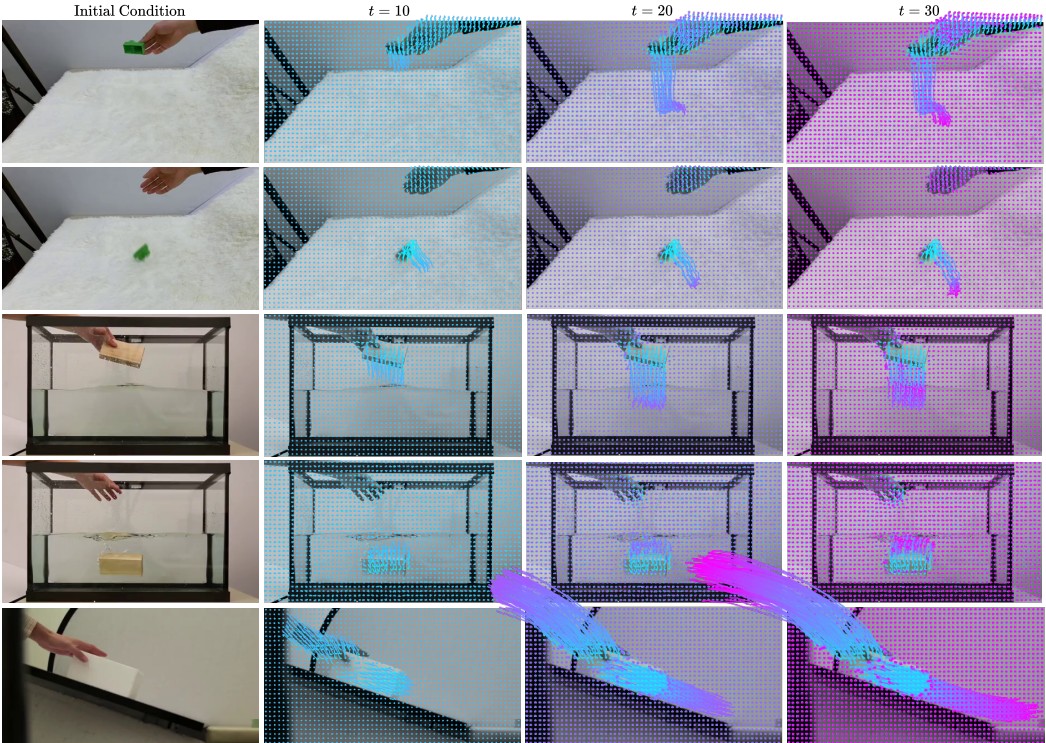

Figure 10: **More qualitative results of our method on Physics101**. The visualization overlays predicted motion trajectories on the initial frame. The color gradient represents temporal progression: blue indicates early timesteps, purple shows intermediate stages, and pink marks later timesteps. Our method can generate the motion of both rigid and non-rigid objects and understand multiple physical phenomena with different material properties (third row and forth row).

**For a clearer understanding of dynamics illustrated in Figures 8 to 10, please refer to the animated visualizations available in our supplementary website materials.**

## A.7 QUALITATIVE RESULTS ON CITYSCAPES

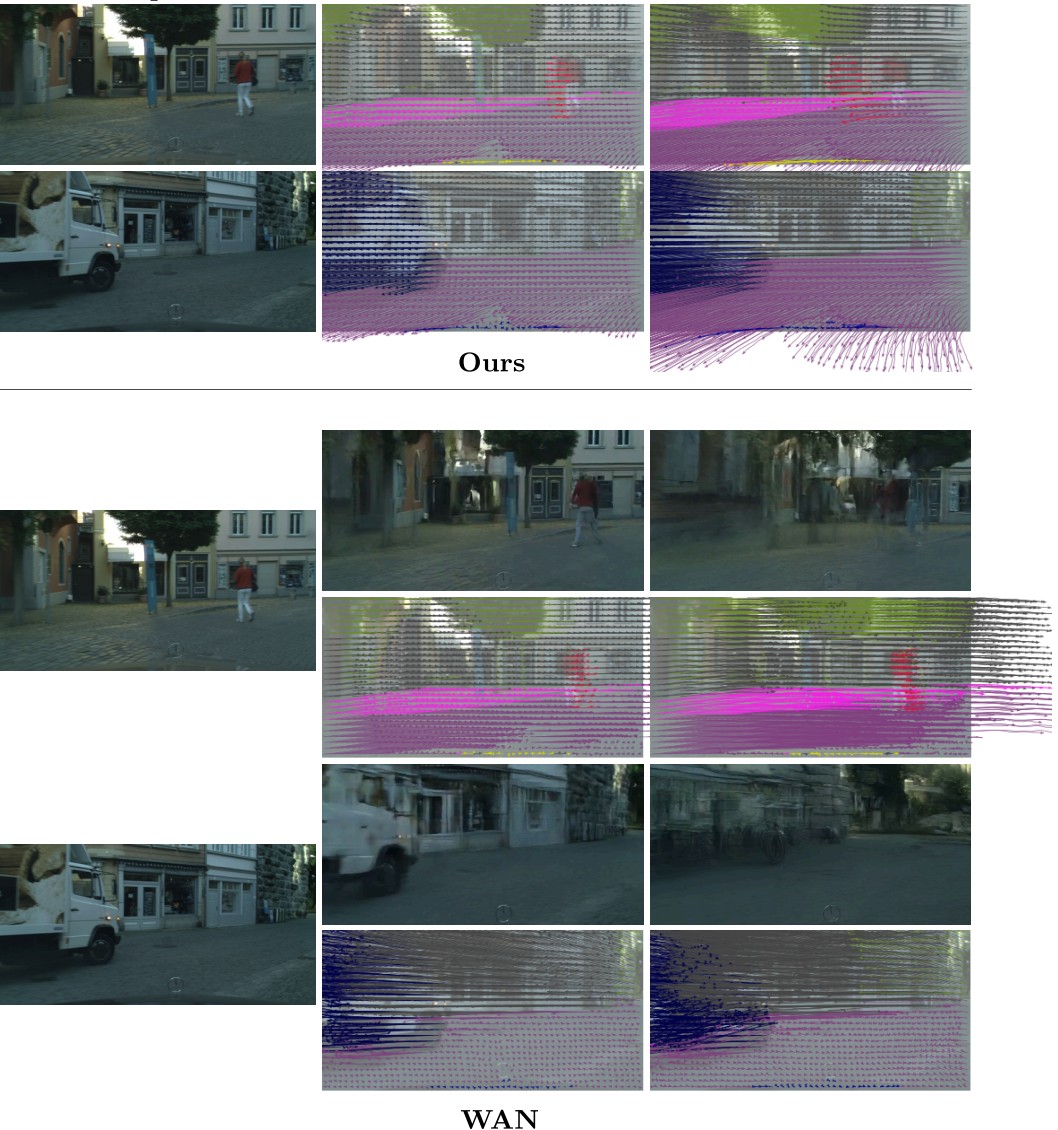

Figure 11: **Qualitative comparison with WAN on Cityscapes**. Our method produces more accurate motion, especially in turning scenarios. WAN frequently hallucinates during turns, generating RGB outputs that confuse the point tracker. In contrast, our method maintains coherent scene-wide motion throughout these challenging conditions, involving large camera and object motion.

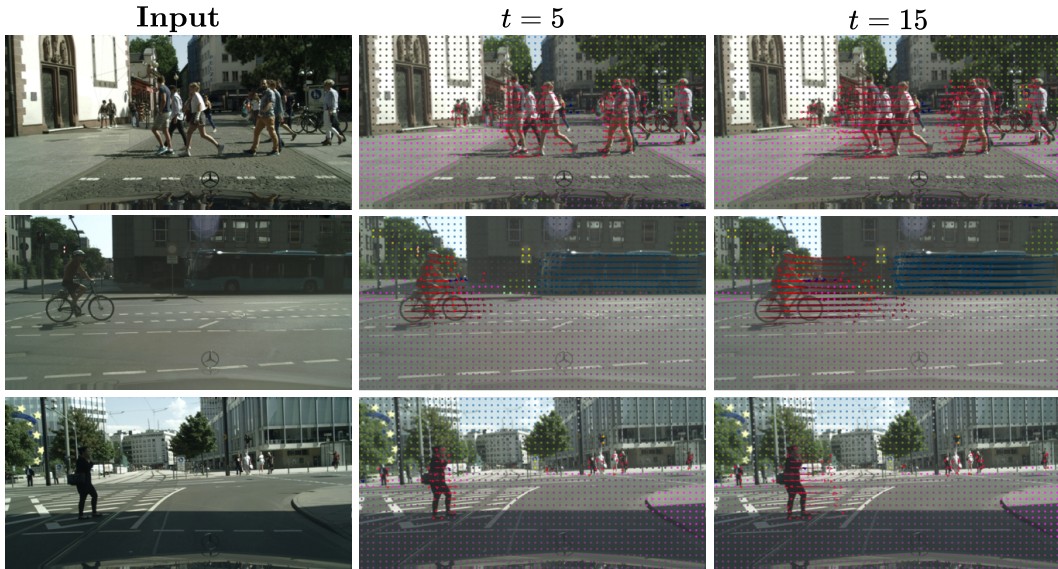

Figure 12: **Predictions of our methods on Cityscapes**. Our method produces accurate motion of non-rigid bodies, such as humans.

## B IMPACT OF DESIGN CHOICES

Table 14: **Ablations of our method on Kubric**: estimated tracks (2), no VAE (4), model sizes (1,3,5).

| Model | FVMD | FVMD (scene) | Best of K | LRTL |
|---|---|---|---|---|
| (1) **Ours (L)** | 13745 | 17838 | 127.0 | 14.6 |
| (2) **Ours (L)** + CT | 14766 | 19123 | 140.3 | 16.4 |
| (3) **Ours (B)** | 12221 | 14950 | 127.2 | 15.9 |
| (4) **Ours (B)** w/o VAE | 14519 | 18551 | 137.8 | 23.7 |
| (5) **Ours (S)** | 13065 | 17119 | 177.6 | 23.6 |

Table 15: **Ablation of VAE** on LIBERO using MSE ($k = 8$). Our method performs better with VAE.

| | LIBERO-90 | | LIBERO-10 | |
|---|---|---|---|---|
| Model | Side | Effector | Side | Effector |
| Ours (B, w/o VAE) (MeanT) | 18.66 | 57.08 | 23.67 | 80.33 |
| Ours (B, w/o VAE) (Mean) | 20.8 | 66.69 | 27.81 | 92.98 |
| Ours (B, w/o VAE) (Min) | 12.55 | 33.17 | 14.99 | 42.47 |
| **Ours (B, VAE) (MeanT)** | 16.70 | 52.70 | 23.69 | 58.35 |
| **Ours (B, VAE) (Mean)** | 18.32 | 60.47 | 26.71 | 66.35 |
| **Ours (B, VAE) (Min)** | 10.99 | 32.01 | 13.86 | 35.93 |

Table 16: **Ablation of VAE** on Physics101 using MSE. Our method overall performs worse with VAE.

| | Physical Scenario | | | | | |
|---|---|---|---|---|---|---|
| Model | Fall | Liquid | Multi | Ramp | Spring | Overall |
| **Ours (w/o VAE)** | 19.78 | 6.00 | 15.65 | 36.35 | 65.31 | 28.62 |
| **Ours (VAE)** | 21.30 | 4.39 | 14.51 | 42.15 | 75.98 | 31.67 |

**Source of trajectories.** We experiment with changing the target distribution for our model on Kubric from ground truth trajectories to those estimated using CoTracker, to support larger-scale, real-world applications, which may rely on estimates of motion. We train both the VAE and the denoiser entirely from scratch using CoTracker trajectories. Comparing (1) vs (2) in Table 14, training on CoTracker output causes a modest performance drop relative to using the actual ground truth. Nevertheless, our method performs comparably or better than the strongest alternative method.

**Model scale.** In Table 14, we also experiment with varying the size of our denoiser between large (L), base (B), and small (S) for (1), (3), (5), respectively. Larger models exhibit less jitter and better geometry preservation, as evidenced by lower LRTL and qualitative results across the evaluation dataset.

**Latent space.** We investigate whether it is necessary to predict trajectories in a latent space with additional downsampling using the VAE. We adjust the patch size such that both the latent flow matching and raw trajectory models process inputs of identical dimensionality. In Table 14, we compare the two settings ((3) and (4)), showing that latent flow matching consistently outperforms flow matching on point coordinates. We further investigate this gap in Figure 13, where we evaluate the *sample variance*[1] of trajectories as the training progresses. Without a VAE, the denoiser collapses to a single mode. Since the ground truth is multi-modal, with infinitely many plausible future

---

[1] Let $\mathbf{X} \in \mathbb{R}^{K \times T \times H \times W \times 2}$ be a tensor of $K$ samples for a given scene (initial image). The *scene sample variance*, denoted $\kappa(\mathbf{X})$, is defined as:

$$\kappa(\mathbf{X}) = \frac{1}{KTHW} \sum_{k=1}^{K}\sum_{t=1}^{T}\sum_{h=1}^{H}\sum_{w=1}^{W} (\mathbf{X}_{k,t,h,w} - \mu_{t,h,w})^2 \,, \text{ where } \mu_{t,h,w} = \frac{1}{K}\sum_{k=1}^{K} \mathbf{X}_{k,t,h,w}.$$

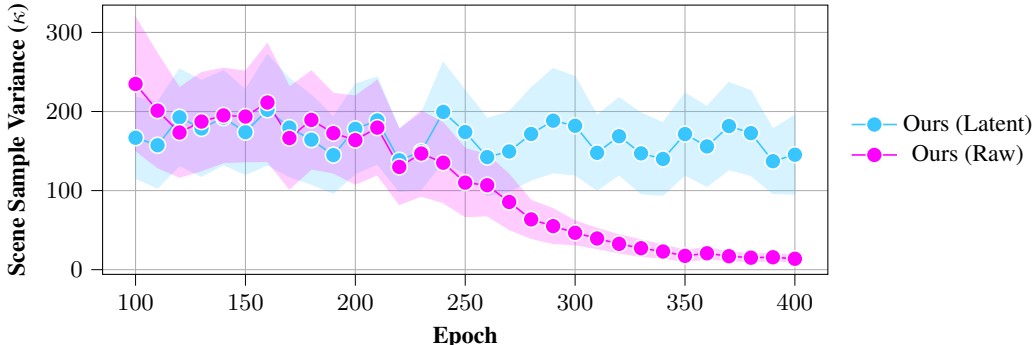

Figure 13: **Scene sample variance** ($\kappa$) on Kubric, shown with one standard deviation around its mean over the dataset. Unlike denoising latent codes, using raw coordinates leads to *single mode collapse*.

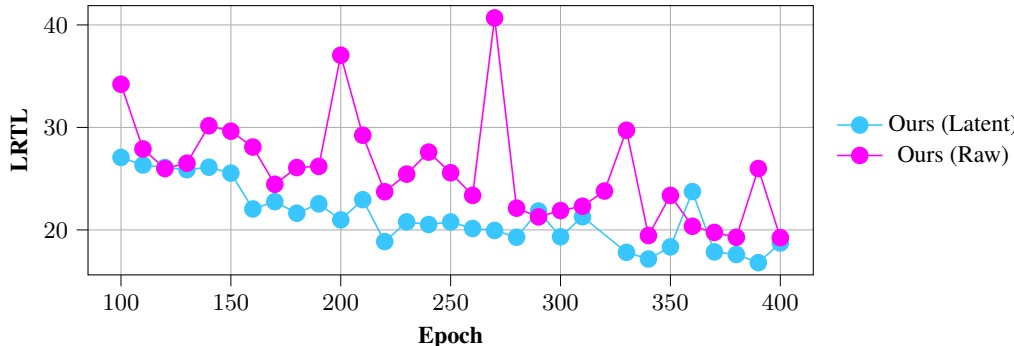

Figure 14: **LRTL through training** on Kubric. Our method produces more plausible motion with VAE.

trajectories, this collapse leads to poor coverage of the target distribution, adversely affecting the distributional metrics and best-of-$K$. We hypothesize that latent modeling is superior because the VAE latent space is smooth and thus easier to model than the raw coordinate space. To qualitatively verify this smoothness, in Figure 15 we show that it is possible to interpolate between distinct sets of plausible ground truth trajectories in the latent space.

Table 15 shows that our method performs better with VAE on robotics data. Table 16 shows that our VAE ablation is less conclusive on real data. Overall, our method performs slightly worse with VAE, but there are $2/5$ scenarios where it performs better. A qualitative inspection suggests that the effect arises from increased diversity, consistent with our observations on Kubric (Figure 13). Since our evaluation on real data is less extensive (only one sample per initial condition), we place greater emphasis on the results obtained from Kubric and LIBERO in the context of our method performance with VAE.

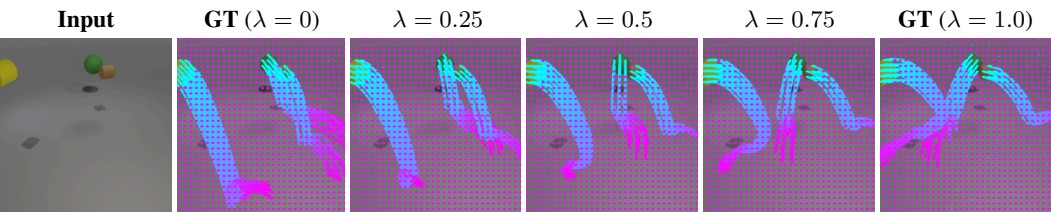

Figure 15: **Decoded latent space interpolations,** $\gamma(\lambda) = (1-\lambda)\mathbf{z}_l + \lambda\mathbf{z}_r$, where $\mathbf{z}_l$ and $\mathbf{z}_r$ are latent codes of two different sets of ground truth future trajectories for the same initial condition (Input).

Table 17: **Effect of sampling density on motion quality.** Denser models consistently outperform sparser ones across all metrics. As grid density increases, the Best-Of-K MSE decreases substantially, demonstrating that modeling more points in the scene produces more accurate motion generation. The best result is shown in **bold**, and the second best is underlined.

| Grid Size | FVMD | FVMD (S) | Best-Of-K | LRTL |
|---|---|---|---|---|
| *Base* | | | | |
| Ours (8×8) | 163.43 | 219.81 | 147.37 | 1.77 |
| Ours (16×16) | 162.29 | 214.00 | 123.08 | 1.54 |
| Ours (32×32) | 155.94 | 187.56 | 78.29 | 1.50 |
| *Large* | | | | |
| Ours (32×32) | **146.28** | **186.92** | **72.75** | **1.20** |

**Sampling density.** We evaluate three grid configurations: 8×8, 16×16, and 32×32. For each configuration, we train a separate VAE using a shared architecture and hyperparameters, varying only the input size (sampling density). All VAEs are trained for 400 epochs, with model selection based on validation set reconstruction error (MSE). We then train separate denoising models using the *base* network architecture, following the input modality ablation (Section 4.3). Each denoiser is trained for 400 epochs with checkpoints saved every 25 epochs. Following the protocol in Section 4.3, we evaluate models using three categories of metrics: distributional (dataset-level and scene-level FVMD), pointwise (Best-Of-K MSE), and data-invariant (LRTL). For each denoiser, we generate samples from the final three checkpoints (epochs 350, 375, and 400) and report the best score among them. To ensure fair comparison, all models are evaluated on a common 8×8 subgrid of points.

Table 17 contains the results. Denser models consistently outperform sparser ones. As grid density increases, Best-Of-K MSE decreases substantially, indicating that modeling more points on the scene leads to more accurate generation. Simultaneously, LRTL decreases with higher density, showing that the generated motion becomes more rigid, better matching the data invariant. This improvement is also evident in the qualitative results. Together, these findings highlight our contribution: *modeling the motion of the entire scene rather than sparse (active) points*.

Since the 32x32 model generates at least 4x more points than the baseline, we also report results for a 32x32 model with a *large* denoiser (roughly 5x more parameters than the *base* version, Table 18a) while maintaining the same VAE. These results demonstrate that scaling the architecture yields further improvements and that generating more points benefits from increased model capacity.

## C   IMPLEMENTATION DETAILS

### C.1   SHARED ARCHITECTURE

The encoder $\phi$, decoder $\psi$, and denoiser $v$ are all based on variants of the same architecture, derived from *Latte* Ma et al. (2025). Originally introduced as a text-to-video denoiser, we adapt it into an image-conditioned spatio-temporal trajectory transformer that functions as either a VAE or a denoiser. The model operates on two types of tokens: trajectory tokens $\mathbf{x} \in \mathbb{R}^{\frac{H}{s} \times \frac{W}{s} \times T \times D}$ and image tokens $\mathbf{f} \in \mathbb{R}^{\frac{H}{p} \times \frac{W}{p} \times D}$ extracted from the image $\mathbf{I}$.

For the VAE, $\mathbf{x}$ is produced by encoding and patchifying the trajectory, as described in Section 3.1. For the denoiser, $\mathbf{x}$ corresponds to rescaled latent codes, $\frac{1}{\gamma}\mathbf{z}$, where $\gamma \in \mathbb{R}^D$ is the per-channel standard deviation computed on the training set. In both cases, image tokens $\mathbf{f}$ are DINOv2 patch features projected to the model dimension via a linear layer. The model alternates between spatial and temporal transformer blocks, folding the corresponding dimension into the batch dimension. To incorporate image context, we extend each *Latte* block with a learnable, gated cross-attention mechanism over the image tokens $\mathbf{f}$. Each block thus consists of self-attention (applied only to trajectory tokens), cross-attention (where trajectory tokens serve as queries and image tokens as keys/values), and a pointwise MLP. After all spatio-temporal blocks, the output is projected and reshaped — either to the latent code shape (if denoising) or to the full trajectory grid (for the VAE).

We consider three different model configurations: small (S), base (B), and large (L).

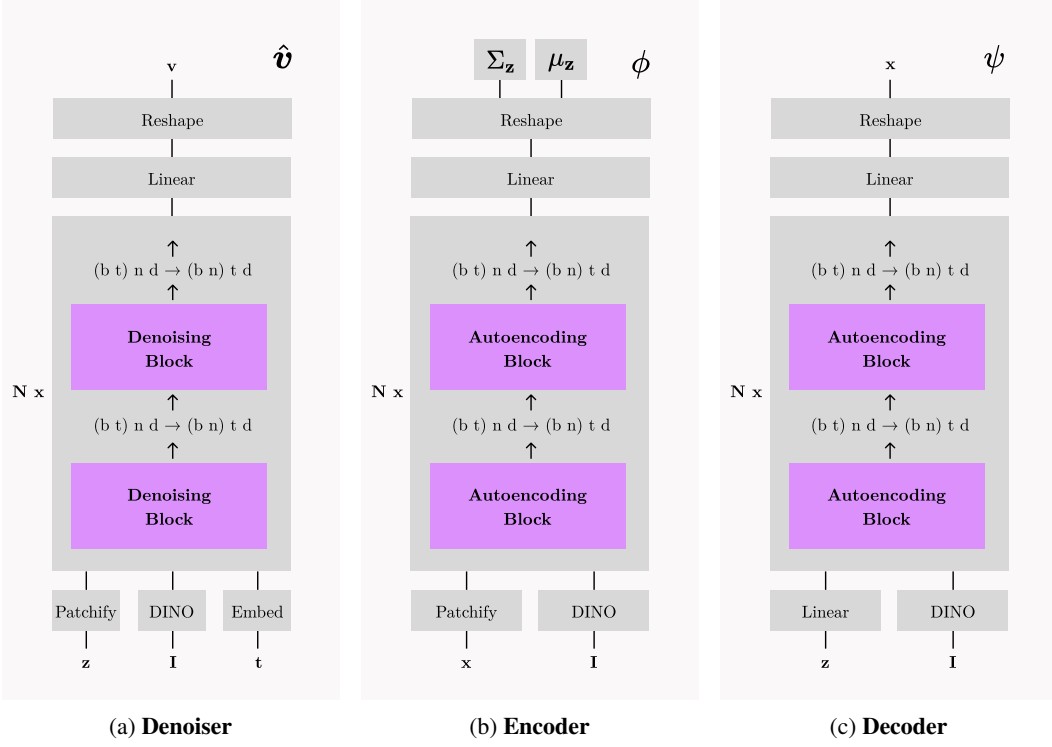

(a) **Denoiser**   (b) **Encoder**   (c) **Decoder**

Figure 16: **Shared Architecture.** The Denoiser $\hat{v}$, Encoder $\phi$, and Decoder $\psi$ all use the same architectural building blocks. The primary difference lies in the type of blocks they alternate: the Denoiser $\hat{v}$ uses Denoising blocks, while the Encoder $\phi$ and Decoder $\psi$ use Autoencoding blocks, illustrated in Fig. 17.

The overall architecture of each component in our method is illustrated in Fig. 16. As previously discussed, our method comprises three neural networks: the **Denoiser** (velocity prediction model, Fig. 16a), the **Encoder** (Fig. 16b), and the **Decoder** (Fig. 16c). All three networks share the same architectural building blocks, alternating between *spatial attention* and *temporal attention* blocks. The inputs to the model are linearly projected to model dimension. Then, they are processed by a

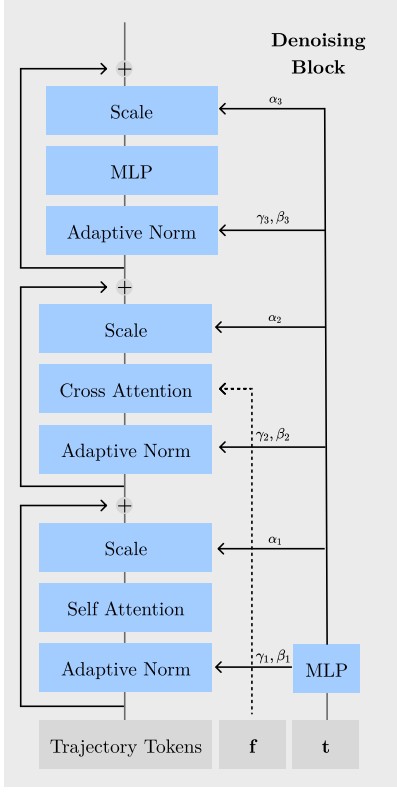
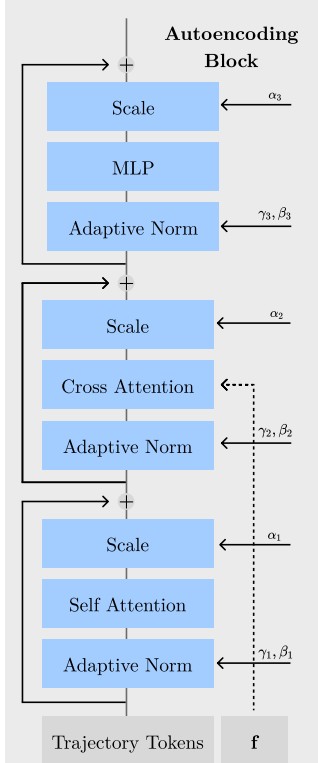

(a) **Denoising Block**    (b) **Autoencoding Block**

Figure 17: **Detailed overview of our attention blocks.** Both the Denoising Block (Fig. 17a) and the Autoencoding Block (Fig. 17b) are adapted from the DiT blocks used in *Latte* Ma et al. (2025). We extend these blocks by introducing image conditioning through gated cross-attention with image features $\mathbf{f}$. In the Denoising Block, a temporal embedding $\mathbf{t}$ is used to predict shift and scale parameters $\alpha$, $\beta$, and $\gamma$ for gating and adaptive normalization. These parameters are predicted by a block-specific MLP. In contrast, the Autoencoding Block uses learnable constants for these parameters.

stack of spatio-temporal blocks. After all spatio-temporal blocks, the output is projected and reshaped, either to the latent code shape (if denoising) or to the full trajectory grid (for the VAE).

In *spatial attention*, the attention mechanism operates across trajectory tokens at a fixed timestep. Conversely, *temporal attention* attends to tokens along the temporal axis within the same trajectory. This is implemented by reshaping the input tensor to fold either the temporal or spatial dimension into the batch dimension, followed by an application of either the **Denoising block** (in the velocity prediction model, Fig. 17a) or the **Autoencoding block** (in the autoencoder, Fig. 17b). Although the most recent works in video generation Wan et al. (2025) use *full attention* (i.e., attention along both spatial and temporal axes), we apply the above-described *factorized attention* due to the quadratic computational cost of *full* attention. In addition, existing point trackers Karaev et al. (2024b;a) demonstrated exceptional accuracy and efficacy of *factorized attention* in point tracking.

Both the **Denoising block** and the **Autoencoding block** are illustrated in Fig. 17, and they follow a shared architectural design. Each block thus consists of self-attention (applied only to trajectory tokens), cross-attention (where trajectory tokens serve as queries and image tokens as keys/values), and a pointwise MLP. Since our task involves predicting trajectories from images, we condition the network on image features $\mathbf{f}$, which are extracted using DINO. In each block, we apply cross-attention between the trajectory tokens and the image features $\mathbf{f}$. The resulting cross-attention output is then combined with the previously computed features through a learnable additive gating. The learnable gating is the addition of previously computed features and scaled gated features, while gating is

pointwise multiplication with the given scale parameter. In the denoising block, the gating parameters are predicted by an MLP, whereas in the autoencoding block, they are learnable constants.

The main distinction between the **Denoising block** and the **Autoencoding block** is due to the additional input used during denoising: a **time embedding**. This embedding conditions the Denoiser on the flow matching timestep, effectively informing the model of the expected noise level in the input. Following the design of *Latte* Ma et al. (2025) and DiT Peebles & Xie (2023), we encode the timestep using sinusoidal positional encoding, followed by a multilayer perceptron (MLP). We condition the denoising model on timestep encoding with *Adaptive Normalization*. We implement *Adaptive Normalization* by first applying *RMSNorm* without elementwise affine parameters and then shifting and scaling the result by the parameters regressed by an MLP. In **Autoencoding block**, we do not have the timestep encoding but we still apply the same normalization and gating. In this case, shift and scale are simply learnable constants. Unlike the original *Latte* Ma et al. (2025), which does not apply QK Normalization, we apply QK Normalization to every attention block in the network. We found this crucial for training stability, consistent with findings from existing works Polyak et al. (2025); Esser et al. (2024); HaCohen et al. (2024b) that apply transformers to flow matching or denoising diffusion. Since the most recent methods Polyak et al. (2025) implement QK Normalization by applying `RMSNorm` to queries and keys, we follow the same method. Thus, we replaced all `LayerNorm` layers with `RMSNorm` to ensure consistent normalization throughout the network.

To summarize, the difference between ours and the *Latte* blocks is that we 1) apply gated cross-attention with image features, 2) apply QK Normalization, 3) use `RMSNorm` instead of `LayerNorm`.

## C.2 Model configurations

We experiment with three different denoising model configurations, shown in Table 18.

Table 18: **Model Configurations**.

(a) **Denoiser Configuration**

| Model | Blocks (N) | Hidden size | Heads | Parameter Count (M) |
|---|---|---|---|---|
| **Small (S)** | 8 | 192 | 3 | 7.2 |
| **Base (B)** | 12 | 384 | 6 | 41.7 |
| **Large (L)** | 16 | 768 | 12 | 220.0 |

(b) **VAE Configuration**

| Model | Encoder Blocks | Decoder Blocks | Hidden size | Heads | Parameter Count (M) |
|---|---|---|---|---|---|
| **Base (B)** | 12 | 12 | 384 | 6 | 57.8 |

For Kubric, we extract image features using DINOv2 (Small), for others using DINOv2 (Large).

## C.3 Model selection

For every dataset, we have reserve a validation set for model selection. For VAEs, we select the best model based on the smallest validation reconstruction loss (L1). For denoisers, model selection criterion is the best validation Best-of-K metric.

## C.4 Training

We use the identical initialization method as *Latte*. We implement all the models in PyTorch and train using Distributed Data Parallel (DDP) with automatic mixed precision in `bfloat16`. For flow matching, we sample timesteps using *logit-normal* method, following MovieGen Polyak et al. (2025). Depending on model size and the dataset, we use different hardware and train for different duration. All the experiments are executed on the internal SLURM compute cluster. Table 19 and Table 20 contain model and training hyperparameters, for each experiment. To reproduce all the experiments in this paper, we estimate total compute time of 32 GPU days. Training resources for each experiment are in Table 21.

Table 19: **Denoiser Hyperparameters**.

(a) **Kubric (S)**

| Hyperparameter | Value / Setting |
|---|---|
| patch size | 1 |
| training epochs | 400 |
| effective batch size | 64 |
| batch size per GPU | 16 |
| gradient accumulation | 1 |
| optimizer | AdamW |
| base learning rate | $6 \times 10^{-5}$ |
| lr scheduler | linear warm-up |
| warm-up steps | 864 |
| clip grad norm | 1.0 |
| latent shape | $24 \times 16 \times 16 \times 8$ |
| track length | 24 |
| sampling method | Euler |
| sampling steps | 10 |
| sampling atol | $1 \times 10^{-7}$ |
| sampling rtol | $1 \times 10^{-7}$ |

(b) **Kubric (B)**

| Hyperparameter | Value / Setting |
|---|---|
| patch size | 1 |
| training epochs | 400 |
| effective batch size | 64 |
| batch size per GPU | 16 |
| gradient accumulation | 2 |
| optimizer | AdamW |
| base learning rate | $6 \times 10^{-5}$ |
| lr scheduler | linear warm-up |
| warm-up steps | 1998 |
| clip grad norm | 1.0 |
| latent shape | $24 \times 16 \times 16 \times 8$ |
| track length | 24 |
| sampling method | Euler |
| sampling steps | 10 |
| sampling atol | $1 \times 10^{-7}$ |
| sampling rtol | $1 \times 10^{-7}$ |

(c) **Kubric (L)**

| Hyperparameter | Value / Setting |
|---|---|
| patch size | 1 |
| training epochs | 400 |
| effective batch size | 32 |
| batch size per GPU | 4 |
| gradient accumulation | 2 |
| optimizer | AdamW |
| base learning rate | $6 \times 10^{-5}$ |
| lr scheduler | linear warm-up |
| warm-up steps | 3238 |
| clip grad norm | 1.0 |
| latent shape | $24 \times 16 \times 16 \times 8$ |
| track length | 24 |
| sampling method | Euler |
| sampling steps | 10 |
| sampling atol | $1 \times 10^{-7}$ |
| sampling rtol | $1 \times 10^{-7}$ |

(d) **LIBERO**

| Hyperparameter | Value / Setting |
|---|---|
| patch size | 1 |
| training epochs | 1500 |
| effective batch size | 64 |
| batch size per GPU | 16 |
| gradient accumulation | 1 |
| optimizer | AdamW |
| base learning rate | $6 \times 10^{-5}$ |
| lr scheduler | linear warm-up |
| warm-up steps | 1448 |
| clip grad norm | 1.0 |
| latent shape | $16 \times 16 \times 16 \times 8$ |
| track length | 16 |
| sampling method | Euler |
| sampling steps | 10 |
| sampling atol | $1 \times 10^{-7}$ |
| sampling rtol | $1 \times 10^{-7}$ |

(e) **Physion**

| Hyperparameter | Value / Setting |
|---|---|
| patch size | 1 |
| training epochs | 40 |
| effective batch size | 32 |
| batch size per GPU | 4 |
| gradient accumulation | 2 |
| optimizer | AdamW |
| base learning rate | $6 \times 10^{-5}$ |
| lr scheduler | linear warm-up |
| warm-up steps | 2048 |
| clip grad norm | 1.0 |
| latent shape | $30 \times 16 \times 16 \times 8$ |
| track length | 30 |
| sampling method | Euler |
| sampling steps | 10 |
| sampling atol | $1 \times 10^{-5}$ |
| sampling rtol | $1 \times 10^{-5}$ |

(f) **Physics101**

| Hyperparameter | Value / Setting |
|---|---|
| patch size | 1 |
| training epochs | 750 |
| effective batch size | 32 |
| batch size per GPU | 4 |
| gradient accumulation | 2 |
| optimizer | AdamW |
| base learning rate | $6 \times 10^{-5}$ |
| lr scheduler | linear warm-up |
| warm-up steps | 2048 |
| clip grad norm | 1.0 |
| latent shape | $30 \times 16 \times 29 \times 8$ |
| track length | 30 |
| sampling method | Euler |
| sampling steps | 10 |
| sampling atol | $1 \times 10^{-5}$ |
| sampling rtol | $1 \times 10^{-5}$ |

## C.5 EVALUATION COST

### C.5.1 KUBRIC

We evaluate all models on 2,048 videos (2 benchmarks (in-distribution, OOD), 64 rollouts, 16 scenes). Evaluation in this field is generally difficult due to the cost of running video generators (sampling for the single initial condition and seed takes more than 2 minutes for most of the baselines; Table 22). In total, our evaluation on Kubric used roughly 500 GPU hours. Note that training our method (L) takes 8.5 days = 204 GPU hours, implying that evaluation is 2x more expensive than training the largest configuration our method.

## D EXPERIMENTAL SETUP

### D.1 COTRACKER'S ACCURACY ON OUR BENCHMARK

We evaluate CoTracker using standard point tracking evaluation (Doersch et al., 2023) metrics and protocol. Table 23 contains the results.

### D.2 ASSESSING METRICS

We investigate the validity of our distributional metrics in Table 24. Here, we make use of our Kubric evaluation set, but we partition it into two sets, such that each scene has 32 possible futures. Intuitively, ground truth data should be a very good predictor of itself. We compare the values against simply running CoTracker to predict the motion of points on the ground truth video. All metrics are lower when ground truth is used to predict ground truth.

Table 20: **VAE Hyperparameters**.

(a) **Kubric**

| Hyperparameter | Value / Setting |
|---|---|
| patch size | 2 |
| training epochs | 400 |
| effective batch size | 16 |
| batch size per GPU | 4 |
| gradient accumulation | 1 |
| optimizer | AdamW |
| base learning rate | $6 \times 10^{-5}$ |
| clip grad norm | 1.0 |
| latent shape | $24 \times 16 \times 16 \times 8$ |
| track length | 24 |
| beta | $1 \times 10^{-6}$ |

(b) **LIBERO**

| Hyperparameter | Value / Setting |
|---|---|
| patch size | 4 |
| training epochs | 400 |
| effective batch size | 16 |
| batch size per GPU | 4 |
| gradient accumulation | 1 |
| optimizer | AdamW |
| base learning rate | $6 \times 10^{-5}$ |
| clip grad norm | 1.0 |
| latent shape | $16 \times 16 \times 16 \times 8$ |
| track length | 16 |
| beta | $1 \times 10^{-6}$ |

(c) **Physion**

| Hyperparameter | Value / Setting |
|---|---|
| patch size | 2 |
| training epochs | 24 |
| effective batch size | 16 |
| batch size per GPU | 4 |
| gradient accumulation | 1 |
| optimizer | AdamW |
| base learning rate | $6 \times 10^{-5}$ |
| clip grad norm | 1.0 |
| latent shape | $30 \times 16 \times 16 \times 8$ |
| track length | 30 |
| beta | $1 \times 10^{-6}$ |

(d) **Physics101**

| Hyperparameter | Value / Setting |
|---|---|
| patch size | 2 |
| training epochs | 400 |
| effective batch size | 16 |
| batch size per GPU | 4 |
| gradient accumulation | 1 |
| optimizer | AdamW |
| base learning rate | $6 \times 10^{-5}$ |
| clip grad norm | 1.0 |
| latent shape | $30 \times 16 \times 29 \times 8$ |
| track length | 30 |
| beta | $1 \times 10^{-6}$ |

Table 21: **Required resources**.

(a) **Denoiser**

| Model | GPUs | Training Time (Days) |
|---|---|---|
| *Kubric* | | |
| **Small (S)** | $4 \times$ A6000 | 3 |
| **Base (B)** | $2 \times$ H100 NVL | 3 |
| **Large (L)** | $4 \times$ H100 NVL | 4.5 |
| *LIBERO* | | |
| **Base (B)** | $2 \times$ H100 NVL | 4 |
| *Physion* | | |
| **Base (B)** | $4 \times$ L40s | 5 |
| *Physics101* | | |
| **Base (B)** | $2 \times$ H100 NVL | 4 |

(b) **VAE**

| Model | GPUs | Training Time (Days) |
|---|---|---|
| *Kubric* | | |
| **Base (B)** | $4 \times$ A6000 | 4 |
| *LIBERO* | | |
| **Base (B)** | $4 \times$ A6000 | 4 |
| *Physion* | | |
| **Base (B)** | $4 \times$ A6000 | 4 |
| *Physics101* | | |
| **Base (B)** | $4 \times$ A6000 | 4 |

## D.3 PROTOCOL FOR VIDEO GENERATORS

**Data Preprocessing:** The Kubric training dataset is originally rendered at 256×256 resolution. However, most video generation baselines assume 16:9 aspect ratio with resolutions such as 320×512 (DynamicCrafter), 320×576 (StableVideoDiffusion), or 640×480 (WAN). To ensure compatibility with pre-training resolution and aspect ratio, we bilinearly upsample Kubric videos such that the shorter side matches the target resolution, and pad the longer side with black bars to preserve the aspect ratio and center the content. For example, for DynamicCrafter, we upsample Kubric to 320×320 and add symmetric vertical padding to produce 320×512 videos.

**Temporal Horizon Adjustment:** As the baselines vary in their output temporal horizon, we modify each implementation to produce 24 frames at 12 fps, ensuring consistent evaluation across models.

**Fine-tuning Protocol:** For DynamicCrafter and WAN2.1, we use the official implementations. For SVD, due to the absence of an official training script, we adopt the Hugging Face version and

Table 22: **Model size and cost.** We compare inference time, memory usage, and parameter count.

| Model | Time (s) | Peak GPU Memory (GB) | Denoiser Size (M) |
|---|---|---|---|
| WAN | 248.2 | 32.3 | 14000 |
| DynamicCrafter[†] | 134.7 | 12.3 | 1487 |
| SVD[†] | 41.8 | 12.7 | 1525 |
| **Ours (L)** | **2.9** | **1.9** | **220** |
| **Ours (B)** | **1.0** | **0.9** | **41.7** |
| **Ours (S)** | **0.5** | **0.8** | **7.2** |

Table 23: **Point-tracking accuracy.** CoTracker3 performs exceptionally on our benchmark data.

| Benchmark | Occlusion Accuracy | Average Jaccard | Avg. Points Within Threshold |
|---|---|---|---|
| Kubric (Out-Distribution) | 91.9 | 84.9 | 91.5 |
| Kubric (In-Distribution) | 91.3 | 82.2 | 90.2 |

implement a custom training pipeline based on publicly available details. We fine-tune each video generator for the same amount of time as it takes to train our method.

**Trajectory Extraction:** We employ the official CoTracker3 implementation to extract trajectories from generated videos. For evaluation, all trajectories are resized to $256 \times 256$ to match the resolution of the ground truth.

### D.4 PROTOCOL FOR TRAJECTORY METHODS

#### D.4.1 REGRESSION METHODS

We compare our method with ATM (Wen et al., 2024a) and Tra-MoE (Yang et al., 2025). To ensure a fair comparison, we carefully design the following training and evaluation protocol.

Using the same dataset and the identical train/validation split as each baseline, we slice videos into 16-frame windows, matching baseline prediction length. For each window, we extract point trajectories at every other pixel (e.g., a $64 \times 64$ grid) using CoTracker. Since both baselines condition trajectory generation on both the initial frame and a text instruction, we extend our method with text conditioning. Specifically, we extract pooled BERT embedding for each text instruction using the same version of BERT as both baselines. Following baselines, we use the same model dimension and first project the instruction embedding with an MLP. Then, we concatenate the projected text embedding with the timestep embedding for flow matching. This is used for conditioning through adaptive normalization, as in the original *Latte* and DiT (Peebles & Xie, 2023). To ensure a fair comparison, we do not apply advanced conditional sampling techniques, such as classifier-free guidance, to our method. We simply train and sample, providing just the input image and the instruction. It is worth noting that this also gives the baselines an advantage due to the experimental observation that unconditional diffusion models are generally worse than the conditional ones in terms of sampling quality (Bao et al., 2022; Li et al., 2024). We compare with each baseline independently, because they are trained on different subsets of the training data. We train our model using raw trajectory grids without any additional preprocessing. For evaluation, we give baselines an advantage by selecting evaluation trajectories using the filtering method they use during training. Specifically, we consider only those trajectories whose temporal variance exceeds a fixed threshold, taken directly from baseline codebase. If there are no such trajectories for a given window, we simply discard the window. Since both baselines predict only 32 trajectories, we first sample uniformly at random 32 trajectories from the set of filtered trajectories. To obtain baseline predictions, we use their official implementation and the checkpoint. For our method, we simply predict trajectories for every other point. Next, from these densely sampled trajectories, we select trajectories corresponding to the query points (i.e. initial positions) of the 32 sampled evaluation trajectories. Because only a single (pseudo) ground-truth trajectory set is available per initial frame, distributional metrics like FVMD or our proposed variants are unsuitable. Instead, we adopt a simple regression metric - mean square error, computing the average Euclidean distance between the $k$ our samples and (pseudo) ground-truth

Table 24: **Verification of distributional metrics.** We check whether true data is a good predictor of itself by partitioning the dataset. We compare this to simply predicting the motion on true videos. The metrics are lowered, showing sensitivity. Note, distributional metrics are sensitive to the number of samples, which here is set to 32.

| Metric | GT vs GT | CoTracker vs GT |
|---|---|---|
| FVMD (scene) | 8979.95 | 21331.69 |
| Best of K | 114.241 | 191.304 |

trajectories. We report results for our method for $k \in \{1, 2, 4, 8\}$. For ATM and Tra-MoE, $k = 1$ always because they are deterministic.

### D.4.2 DIFFUSION-BASED METHODS

Firstly, we briefly introduce the Track2Act (Bharadhwaj et al., 2024b). The method builds on the original diffusion transformer (DiT) (Peebles & Xie, 2023) to model trajectory generation with two-frame conditioning, namely the start image and the goal image. Both a start and a goal frame are encoded using a pre-trained ResNet18, producing one embedding vector per frame. These embeddings are concatenated, flattened, and linearly projected to the model dimension, then injected into the network via adaptive normalization layers to guide generation. The model is trained on variable-length tracks (100–400 frames). The prediction horizon is 8 future frames. Each track is flattened along the channel dimension. The method uses original DDPM (Ho et al., 2020) formulation with 1000 timesteps. Here are the modifications applied to their method:

1. **Dropping the goal frame.** We simply drop the goal frame and use only start frame embedding vector for conditioning.

2. **Extending prediction horizon.** Track2Act originally predicts the next eight frames. We extend the prediction horizon to 24 frames to align with our Kubric training and evaluation setup.

3. **Fixing the number of generated points.** Because Track2Act is built on a DiT (Peebles & Xie, 2023), and transformers are known to generalize poorly to out-of-distribution sequence lengths (Li et al., 2025; Veličković et al., 2025), we standardize the number of generated points. Recall that our method predicts 1,024 points arranged on a $32 \times 32$ grid, whereas the original Track2Act is trained to output at most 400 uniformly sampled points. For a fair comparison, we therefore train Track2Act to produce exactly 1,024 points with the same $32 \times 32$ grid arrangement used by our method and in evaluation.

We train the method on our dataset using the official publicly available codebase, following the default hyperparameter settings provided by the authors.

### D.4.3 PHYSICS101

The dataset consists of roughly 10000 video clips containing 101 objects of various materials and appearances (shapes, colors, and sizes). Since this dataset was collected using high-resolution camera (1080p), resulting in videos where majority of the objects are small compared to the background, we preprocessed the dataset such that the objects and their interactions are centered, preserving the aspect ratio but reducing the resolution to 256x464, for computational reasons. There are five different physical scenarios, namely fall, liquid, multi, ramp and spring. Please see the dataset for more information about these. For computational reasons, we extracted 32x58 trajectory grids using CoTracker3. All the clips consist of 30 frames, 1856 trajectories. Our training set contains 9252 different initial conditions with the single ground truth. Our test set contains 1450 different initial conditions and single ground truth per initial condition.

## E USER STUDY

We carried out a user study comparing our model, fine-tuned SVD and fine-tuned WAN1.3 model. The study consists of 16 questions, asking the respondents to rank the three models from best to

worst in each question. We did not identify the models in the study, but randomly gave them a label as "Option 1", "Option 2", and "Option 3" for each question. We show example of the question in Fig. 18. All questions are identical except the scene animation changes. We include all scenes alongside this supplemental document.

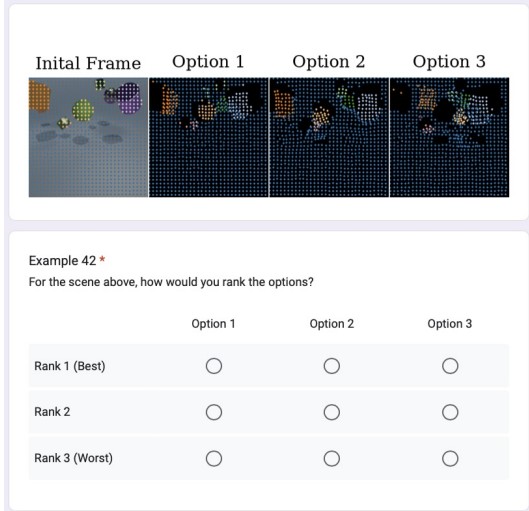

Figure 18: Example of user study question. The study contained 16 questions showing the animation of 3 methods, which were assigned names at random for each question.

## F    More related work

**Studies on the implausibility of motion in generated videos.**    Several studies have highlighted physical implausibility in video generation. To quantify this, Motamed et al. (2025) introduced Physics-IQ, a novel benchmark dataset evaluating the physical understanding of video generation models. Their findings reveal that, while current models exhibit impressive visual realism, their understanding of fundamental physical principles remains limited. Errors include the spontaneous (dis)appearance of objects and physically implausible object interactions (e.g., objects passing through each other). In a related study, Kang et al. (2024) investigated whether video generation using latent diffusion can learn solid mechanics from a simple 2D dataset governed primarily by rigid body mechanics. Their results suggest that scaling up model size improves performance within the training distribution and aids combinatorial generalization but does not lead to accurate motion synthesis out-of-distribution. VideoPhy Bansal et al. (2024) conducted a large-scale user study assessing whether generated videos followed physical common sense, observing limited performance even for the latest video generators.

## G    Ethics Statement and Broader Impact

Our work offers a computationally efficient alternative for inferring motion. This approach has the potential to significantly reduce the resource demands of motion forecasting, making it more accessible and deployable in real-world scenarios, especially on edge devices or in bandwidth-constrained environments. By eliminating the need for video input and the additional processing required for video-based tracking, our model can democratize access to future motion understanding, benefiting fields such as robotics, autonomous navigation, assistive technology, and video editing.

However, the ability to infer motion dynamics from a static image raises ethical considerations. In particular, if used in surveillance or behavioral prediction, such technology could be misapplied to infer intentions or future movements of individuals without their knowledge or consent. These concerns underline the importance of deploying such technologies transparently, with safeguards to protect privacy and civil liberties.

Moreover, the model's reliance on learned priors from training datasets may introduce biases. Future work should explore robustness and domain adaptation to ensure that the benefits of this research extend across diverse contexts.

## H   USE OF LLMS IN OUR WORK

We use LLMs to refine and rephrase our text, as well as to assist in generating visualizations, which are included both in the main body of the paper and in the supplementary material.

## I   LIMITATIONS

Due to computational resource constraints, our work is limited to synthetic data and small-scale real-world experiments. Moreover, lower conditioning image resolution and further downsampling applied by patchification in the encoder reduce the accuracy of predicted tracks around object boundaries for our model. Additionally, while our method demonstrates strong performance on datasets like Kubric, LIBERO, Physics101, and also some generalisation to real-world scenarios from Physion, it is currently limited from achieving broader in-the-wild generalisation. Future efforts will focus on addressing these limitations by scaling up datasets and improving the resolution of the image features.

