# OpenReview forum: "What Happens Next? Anticipating Future Motion by Generating Point Trajectories"
_ICLR.cc/2026/Conference — ICLR 2026 Poster_

### Official Review · Reviewer_GBJN · 2025-10-30

**Soundness:** 4
**Presentation:** 4
**Contribution:** 3
**Rating:** 8
**Confidence:** 4

**Summary:**

In this paper, the authors focus on efficient approaches to forecasting dynamics in an image. To this end, they propose, instead of modeling pixels directly, to forecast point trajectories. They utilize a beta-VAE combined with Flow Matching to generate future point trajectories conditioned on an image. The authors then comprehensively evaluate the performance of their model, looking at both statistical and physical plausibility. They evaluate on robotic datasets such as Libero as well as synthetic datasets (Kubric). The also include a user study of performance.

**Strengths:**

1. The paper explores a less studied avenue of visual dynamics: predicting pixel motion versus pixels themselves. This approach makes intuitive sense.

2. Quantitative evaluation is comprehensive and convincing.

3. The paper is well-written and easy to understand.

**Weaknesses:**

1. There is a prior work, "An Uncertain Future: Forecasting from Static Images using Variational Autoencoders", Walker et al, ECCV 2016, which is similar and also uses a VAE to forecast pixel trajectories from a single image. This work should be cited.

2. Results of the user study are somewhat weak (preferred over other approaches only 52% of the time).

3. It would be helpful to see a few more qualitative results from the Physics 101 dataset.

**Questions:**

1. Please cite the paper above. It is similar to approach mentioned in this paper.

2. Please elaborate more on the user study, and would it be possible to include more results from Physics 101?

---

> ### Author Response · Authors · 2025-11-25
>
> Thank you for your positive review and for recognizing our contributions. We will address your concerns, each in a separate comment due to character limit.
>
>
> > There is a prior work, "An Uncertain Future: Forecasting from Static Images using Variational Autoencoders", Walker et al, ECCV 2016, which is similar and also uses a VAE to forecast pixel trajectories from a single image. This work should be cited.
> > Please cite the paper above. It is similar to approach mentioned in this paper.
>
> Thank you for the reference! *An Uncertain Future: Forecasting from Static Images using Variational Autoencoders* predicts point trajectories from a single image, matching our setting. However, while both methods address the same problem, our study differ fundamentally in three key aspects: (1) motion representation, (2) generative modeling, and (3) evaluation of predicting multiple sensible futures.
>
> **Motion representation.** The above method encodes trajectories as offsets relative to the first frame, then applies a discrete cosine transform (DCT) for compression, downsampling 12× by truncating DCT basis coefficients. This results in a linear compression with no regularization on the latent space structure. In contrast, we tailor our motion representation specifically for generation by training a spatio-temporal VAE. Our approach provides nonlinear compression regularized to match a Gaussian prior, making it better suited for generative tasks.
>
> **Generative modeling.** While both methods employ VAEs, their roles differ significantly. The cited work uses a VAE to directly generate future point trajectories. We instead use a VAE to construct a regularized latent space for generative modeling with rectified flow.
>
> **Multiple futures.** Since extrapolating future from a single image is inherently ambigious, in our study we measure whether multiple futures our model can sample are sensible. We consider whether rigidity invariant is preserved and measure distance between distributions of sampled and true futures for each image. This expands above evaluation performed in prior work. Unlike prior work, we also rigorously evaluate generalization to out-of-distribution object shapes.
>
> We have added this discussion and the citation to the Related Work section of our paper.

---

> ### Author Response · Authors · 2025-11-25
>
> > Results of the user study are somewhat weak (preferred over other approaches only 52% of the time).
>
> We find that assessing motion quality is challenging for human evaluators. This is especially true as the actual underlying video is not available. For a clearer understanding of the evaluation difficulty and the nature of the comparisons, please take a look at the user study scenes already provided in the supplementary material. It is important to note that our user study involves three models, with participants selecting the best among them. Therefore, random chance would be 33%, making our 52% preference rate considerably higher than chance or other methods (28% or 20%).

---

> ### Author Response · Authors · 2025-11-25
>
> > Please elaborate more on the user study, ...
>
> Additional details about the user study are available in the supplementary PDF. Here, we briefly summarise key details. There were 16 questions where the respondeds were asked to rank three models in order. The models were randomly shuffled for each question. We include the actual scenes (the gifs shown to users during the study) in the supplementary archive. Let us know if there are any specific details we may have missed.

---

> ### Author Response · Authors · 2025-11-25
>
> > It would be helpful to see a few more qualitative results from the Physics 101 dataset.
>
> We expanded the supplementary PDF with two pages of qualitative results (Appendix A.6).

---

### Official Review · Reviewer_VFf8 · 2025-11-03

**Soundness:** 3
**Presentation:** 3
**Contribution:** 2
**Rating:** 6
**Confidence:** 3

**Summary:**

This paper addresses the problem of predicting object movement from a single static image, where future motion is inherently uncertain. The authors propose a method that directly generates a dense grid of point trajectories (coordinates) conditioned on the input image, rather than generating future video frames (pixels). They utilize a generative architecture (similar to modern video generators, specifically using flow matching) to model the distribution of possible future movements.

**Strengths:**

1. Generates point trajectories instead of pixels, bypassing the complexity of video generation. This allows the model to efficiently learn physical dynamics and object consistency.
2. Uses a generative model to predict the distribution of trajectories, producing multiple physically plausible futures, which better reflects real-world information gaps.
3. Achieves superior performance in physical reasoning tasks compared to prior trajectory regressors and state-of-the-art video generators pre-trained on massive datasets.

**Weaknesses:**

1. Predictions are based on a single image, lacking precise initial velocity or force data. Accuracy heavily relies on the model's learned "physical common sense" rather than precise observation. I think this task setting quite strange.
2. Evaluation primarily uses synthetic datasets (e.g., Kubric, Physion). Its generalization capability to complex, "in-the-wild" real-world physics (e.g., fluid, soft bodies) remains questionable.

**Questions:**

1. The point I understand the least is why, for a scenario where object motion needs to be predicted, it is limited to using only a single static image. Alternatively, is there a way to adapt this model to also support video or multi-frame input? And how would its performance be?
2. It seems to lack some relevant paper citations, such as Flow-grounded spatial-temporal video prediction from still images, papers about future frame synthesis/prediction

---

> ### Author Response · Authors · 2025-11-25
>
> Thank you for your positive review and thoughtful feedback. We address your concerns below, each in a different comment due to character limit.
>
> > The point I understand the least is why, for a scenario where object motion needs to be predicted, it is limited to using only a single static image.
>
> While our method is not limited to single static images (see the next answer for details), we do focus on generation from single static image. We **intentionally** focus on the most challenging and ambiguous setting: generating diverse, plausible motion from a single frame, without access to auxiliary information such as object velocities, applied forces, or extended temporal context (e.g., multiple frames). Our goal is to evaluate whether a generative model can infer the movable and static components of a scene from one image and then propose diverse, plausible hypotheses for how that scene might evolve.
>
> This is inherently difficult because the model must account for the uncertainty in various physical properties (e.g. velocity) and infer the existence of **multiple possible futures** purely from nearby training examples. However, simple interpolation between training examples is not sufficient or physically plausible. For instance, interpolating two rigid motions does not generally yield a valid rigid motion. This challenge is discussed explicitly in the paper (Lines 224–230). For that reason, we evaluate not only distributional alignment (FVMD), but also pointwise accuracy (Best-Of-K) and physical plausibility (rigidity), as detailed in Section 3.4.
>
> It is also worth noting that predicting motion from static frames, optionally with auxiliary information such as textual prompts, is common in prior robotics literature (e.g., ATM, Tra-MoE, Track-2-Act). Our experiments show that our proposed method outperforms these approaches, both **with** additional conditioning (Tables 1 and 2) and **without** it (Tables 3 and 4).
>
> Finally, trajectory generation from single frame has direct applications in image-to-video generation and video-to-video motion transfer. For instance, recent work ^[1] demonstrates impressive results in object manipulation, camera control, and motion transfer by generating videos conditioned on a **single frame**, **future point trajectories**, and a text prompt. However, this approach assumes future point trajectories are already available, without addressing how to generate them. Our method fills this gap by providing a method for trajectory generation, accompanied by comprehensive evaluation of generated motion quality.
>
> ## References
> [1] [MotionPrompting](https://arxiv.org/abs/2412.02700)

---

> ### Author Response · Authors · 2025-11-25
>
> > Accuracy heavily relies on the model's learned "physical common sense" rather than precise observation. I think this task setting quite strange.
>
> > Alternatively, is there a way to adapt this model to also support video or multi-frame input?
>
> > And how would its performance be?
>
> Our model can be easily extended to support video or multi-frame inputs. Currently, we condition on the initial frame using cross-attention over tokens extracted from the initial frame. To incorporate multiple frames, the model would simply cross-attend to tokens from all conditioning frames, requiring no major architectural changes or training setup changes. Please see Section C.1 of the Appendix for details. To demonstrate this, here we train and evaluate a multi-frame version of our method on Kubric.
>
> **Theoretical formulation.** We train a single model that can generate future trajectories from a variable number of input images. Formally, the multi-frame model learns conditional distributions of future trajectories given a variable number of input images,
> $ (p (\mathbf{X} | \mathbf{I}_1 \ldots \mathbf{I}_k))_1^K  $, where $\mathbf{X} \in \mathbb{R}^{T \times \frac{H}{s} \times \frac{W}{s} \times 2}$ are future trajectories on a spatial grid and $K \in \mathbb{N}$ denotes the maximum context length.
>
> **Implementation.** We cross-attend with tokens from all conditioning frames. To encode temporal ordering of the input frames, we add learnable frame index positional encodings to the frame patch tokens before cross-attention. During training, we randomly vary the context length. The VAE remains unchanged, conditioned only on the initial frame $I_1$.
>
> **Evaluation**
> We evaluate the same model with either 1 or 2 input frames on identical test data. The test set is the same as in the paper, including 16 scenes each for in-distribution and out-of-distribution object shapes, with 64 rollouts per scene. Each rollout shares the same initial object placement but varies in initial velocity. Single-frame prediction is highly ambiguous due to unknown initial velocity. With two frames, velocity is observable and the Kubric data generation process is actually deterministic. However, occlusions still introduce some ambiguity in predictions. Therefore, we sample 3 predictions per input configuration (i.e. number of input frames) and report the mean squared error (MSE) between generated and ground truth trajectories. Here, we follow the evaluation protocol from Section 4.1 since the Kubric data generation process becomes deterministic when velocity is observed.
>
> **Results**
>
>
> |  | MSE(MeanT) | MSE (Mean) | MSE (Min) |
> |----------------:|----------:|---------:|--------:|
> | *Kubric (In-Distribution)* | | | |
> | 1 input frame | 403.287 | 478.369 | 368.784 |
> | 2 input frames| 253.263 | 264.963 | 226.989 |
> | *Kubric (Out-Distribution)* | | | |
> | 1 input frame | 275.284 | 305.264 | 255.165 |
> | 2 input frames | 219.097 | 233.268 | 209.901 |
>
> The multi-frame samples outperform the single-frame samples across all metrics and both dataset splits. Qualitatively, multi-frame predictions are more accurate and exhibit lower diversity, reflecting reduced uncertainty when velocity is observed. Notably, the model adapts its prediction diversity based on the amount of conditioning information provided. This also demonstrates that generation from single frame is indeed more challenging and that our method is capable of incorporating more given information. Given two frames, model relies on learned "physical common sense" and precise observation. Please see additional qualitative evidence in the Appendix A.2 of the updated version of the paper.
>
> We have added these results to the Appendix A.2 of our paper.

---

> ### Author Response · Authors · 2025-11-25
>
> > It seems to lack some relevant paper citations, such as Flow-grounded spatial-temporal video prediction from still images, papers about future frame synthesis/prediction
>
> Thank you for the reference! *"Flow-grounded spatial-temporal video prediction from still images"* is indeed related to our work because it also considers generating motion from the single still frame, but for a different purpose. While both methods address the same problem, our study differ fundamentally in four key aspects: (1) final objective, (2) motion representation, (3) generative modeling, and (4) evaluation of predicting multiple sensible futures.
>
> **Final objective.** The cited paper focuses on image-to-video generation as a two-phase process: flow prediction followed by flow-to-frame generation. In contrast, we focus on motion generation and evaluation. Furthermore, Section 4.3 demonstrates that RGB is a suboptimal modality for reasoning about motion.
>
> **Motion representation.** The cited method represents motion as a sequence of optical flow maps, which has several limitations: it cannot guarantee long-term temporal consistency due to independent frame-pair estimation (causing error accumulation), and it fails under occlusion. To address these limitations, we use point trajectories instead of optical flow. This representation offers remarkable flexibility: it can track any number of points (sparse or dense), capture both object-specific movements and scene-wide motion, and is more robust to occlusion. Also, state-of-the-art point trackers like CoTracker and AllTracker estimate trajectories jointly, ensuring better temporal consistency and tracking through occlusions. We also adopt this approach (lines 47-52). Moreover, we tailor our representation for generation by training a spatio-temporal VAE that performs nonlinear compression regularized to match a Gaussian prior.
>
> **Generative modeling.** While both methods employ VAEs, their roles differ. The cited work uses a VAE to directly generate future flow maps, whereas we use a VAE to construct a regularized latent space for generative modeling with rectified flow.
>
> **Multiple futures.** Since extrapolating the future from a single image is inherently ambiguous, we rigorously measure whether the multiple sampled futures are sensible. We evaluate whether rigidity invariants are preserved and measure distributional distances between sampled and ground-truth futures for each image. This is considerably more extensive than the prior work's RMSE evaluation. We also rigorously evaluate generalization to out-of-distribution object shapes.
>
> We have added this discussion and citation to the Related Work section.

---

### Official Review · Reviewer_h2xU · 2025-11-03

**Soundness:** 2
**Presentation:** 3
**Contribution:** 2
**Rating:** 6
**Confidence:** 3

**Summary:**

This paper addresses the problem of forecasting object motion from a single image—predicting potential object movements without observing velocities or applied forces. It formulates the task as conditional generation of dense point trajectory grids, drawing inspiration from modern video generator architectures but outputting trajectories instead of pixels. The model uses a trajectory VAE for latent space mapping and flow matching for sampling, capturing scene-wide dynamics and uncertainty.

**Strengths:**

1. It innovatively redefines motion forecasting as dense trajectory grid generation, moving beyond prior work focused on sparse active points (e.g., ATM, Tra-MoE) or pixel-based video generation, enabling better global scene dynamics modeling.
2. The study uses diverse datasets and comprehensive metrics to assess accuracy, diversity, and physical plausibility, with ablations strengthening conclusions

**Weaknesses:**

- **Lack of Downstream Task Evaluation for Video Generation**: While the paper argues that trajectory generation outperforms video generation for motion forecasting, it does not test how the two approaches perform on downstream tasks that rely on motion, e.g., robot manipulation, video interpolation, or action prediction.
- **No Analysis of Query Point Sampling Density**: The paper only discusses ablations of latent code dimensions but does not explore the impact of query point sampling density.
- **Insufficient Real-World Data Results and Visualizations**: Real-world validation is limited to the Physics101 dataset without testing on other real datasets, e.g., egocentric motion datasets.

**Questions:**

In the absence of a task description (e.g., for the action "Pick up the book on the right and place it under the cabinet shelf"), how does the paper address the diversity of action prediction, and are there any controllable settings integrated into the pipeline?

---

> ### Author Response · Authors · 2025-11-25
>
> Thank you for the positive review and constructive feedback. We appreciate your comments and will aim to address all your concerns during the rebuttal period. However, some of your points require additional experiments, so we focus on answering your specific question for now, while we work on those results.
>
> > In the absence of a task description (e.g., for the action "Pick up the book on the right and place it under the cabinet shelf"), how does the paper address the diversity of action prediction, and are there any controllable settings integrated into the pipeline?
>
> Our approach intentionally focuses on the most challenging scenario: generating diverse and accurate motion without any controllable settings. This setting allows us to evaluate the model's ability to predict diverse but plausible motion in unconstrained contexts.
>
> **To address diversity with controllable settings**, we have extended our framework to incorporate text prompts as conditioning inputs and demonstrated its applicability on robotics data. When textual prompts are introduced, the model's behavior changes: on robotics data, the generated samples converge around the mean prediction rather than exhibiting unnecessary diversity. This behavior is evident in Figure 3 and is further documented in our supplementary material. Please see samples on robotics data in the sample viewer (offline website) provided with our supplementary material, which demonstrates how text conditioning enables controlled, task-specific motion generation while maintaining the model's capacity for diverse predictions.
>
> Another approach to address prediction diversity is conditioning on more inputs (e.g. several images instead of one). As detailed in our response to reviewer VFf8, our model can be easily extended to support multi-frame or video inputs. We demonstrate that the number of input frames directly controls sample diversity. Specifically, providing multiple frames enables estimation of the initial velocity, consequently reducing trajectory diversity and demonstrating that the model correctly incorporates additional context. Samples generated from more inputs concentrate around the mean, indicating reduced uncertainty in predictions. Notably, this is achieved without more advanced guidance strategies such as classifier-free guidance. This discussion, along with qualitative and quantitative results, is also included in Appendix A.2 of the paper.

---

> ### Author Response · Authors · 2025-11-29
> **Implemented Analysis of Query Point Sampling Density**
>
> Thank you for the patience while we prepared the additional experiments with our limited compute resources (Table 20).
>
>
> > No Analysis of Query Point Sampling Density: The paper only discusses ablations of latent code dimensions but does not explore the impact of query point sampling density.
>
>
> Thank you for suggesting this ablation! We originally omitted it due to its computational cost (Table 19). In our experiments, we simply used the densest configuration that our hardware allows (32×32).
>
> **Implementation.** We evaluate three grid configurations: 8×8, 16×16, and 32×32. For each configuration, we train a separate VAE using a shared architecture and hyperparameters, varying only the input size (sampling density). All VAEs are trained for 400 epochs, with model selection based on validation set reconstruction error (MSE). We then train separate denoising models using the *base* network architecture, following the input modality ablation (Section 4.3). Each denoiser is trained for 400 epochs with checkpoints saved every 25 epochs.
>
> **Evaluation.** Following the protocol in Section 4.3, we evaluate models using three categories of metrics: distributional (dataset-level and scene-level FVMD), pointwise (Best-Of-K MSE), and data-invariant (LRTL). For each denoiser, we generate samples from the final three checkpoints (epochs 350, 375, and 400) and report the best score among them. To ensure fair comparison, all models are evaluated on a common 8×8 subgrid of points.
>
> **Results.**
>
> | **Grid Size** | **FVMD** | **FVMD (S)** | **Best-Of-K** | **LRTL** |
> |---------------|----------|--------------|---------------|----------|
> | *Base*        |          |              |               |          |
> | Ours (8×8)    | 163.43   | 219.81       | 147.37        | 1.77     |
> | Ours (16×16)  | 162.29   | 214.00       | 123.08        | 1.54     |
> | Ours (32×32)  | *155.94* | *187.56* | *78.29* | *1.50* |
> | *Large*       |          |              |               |          |
> | Ours (32×32)  | **146.28** | **186.92** | **72.75** | **1.20** |
>
> The table above reports the results; the best is in **bold**, and the second-best in *italic*. Denser models consistently outperform sparser ones. As grid density increases, Best-Of-K MSE decreases substantially, indicating that modeling more points on the scene leads to more accurate generation. Simultaneously, LRTL decreases with higher density, showing that the generated motion becomes more rigid, better matching the data invariant. This improvement is also evident in the qualitative results. Together, these findings highlight the key benefit of our contribution: *modeling the motion of the entire scene rather than sparse (active) points*.
>
> Since the 32x32 model generates at least 4x more points than the baseline, we also report results for a 32x32 model with a *large* denoiser (roughly 5x more parameters than the *base* version) while maintaining the same VAE. These results demonstrate that scaling the architecture yields further improvements and that generating more points benefits from increased model capacity.
>
> We included this ablation in Appendix Section B, which also contains other ablations and design choice studies.

---

> ### Author Response · Authors · 2025-11-29
> **Downstream Task Evaluation**
>
> Thank you for the patience while we prepared the additional experiments with our limited compute resources (Table 20).
>
> > Lack of Downstream Task Evaluation for Video Generation: While the paper argues that trajectory generation outperforms video generation for motion forecasting, it does not test how the two approaches perform on downstream tasks that rely on motion, e.g., robot manipulation, video interpolation, or action prediction.
>
> Further evaluation on robot manipulation, such as training control policies, falls outside the scope of our work. Our primary focus is on motion generation and evaluation. Tra-MoE, a prior work, has already demonstrated that downstream manipulation success rates are positively correlated with trajectory forecasting accuracy, as shown in Figure 5 of their paper.
>
> For downstream use, we explore image-to-video generation and present preliminary evidence that our trajectory generation method can be used to enhance motion quality in synthesized videos.
>
> **Implementation** We adopt a fixed *base* denoising architecture while varying the input modality, as in Section 4.3. For video generation, we select the StableDiffusion3.5 (SD3.5) RGB latent space, which demonstrates superior motion quality compared to SVD and WAN (Table 6). We train a specialized VAE that encodes 32×32 trajectory grids conditioned on 128×128 images. This trajectory VAE is the same as our 256×256 VAE but operates at lower resolution to align the latent tensor dimensions with the RGB VAE.
>
> We then train two generative models: (1) an RGB-only model that generates video from the SD3.5 latent space, and (2) a joint model that simultaneously generates video and trajectories from both the SD3.5 RGB latent space and the trajectory VAE latent space. Architecturally, we maintain an identical denoising network across both models, varying only the input latent representation. The RGB-only model generates 16-channel SD3.5 latents, while the joint model generates 24-channel inputs latents by concatenating RGB latents (16 channels) with trajectory latents (8 channels). This concatenation is sensible because trajectory VAE grid structure matches RGB VAE grid structure.
>
> **Evaluation** We follow the same evaluation protocol as Section 4.3 for video methods. For both models, we extract a 32×32 grid of trajectories from the generated RGB frames using CoTracker 3. In the RGB+Trajectory model, we discard the directly generated trajectories to ensure a fair comparison. This evaluation approach measures the motion quality of the generated RGB frames themselves, rather than the explicitly generated trajectories. Note that Table 6 already demonstrates that directly generating motion produces superior quality compared to extracting motion from generated RGB frames.
>
> **Results**
>
> |  | Latent Shape | FVMD | FVMD (S) | Best-Of-K | LRTL |
> | :--- | :--- | :--- | :--- | :--- | :--- |
> | *Kubric (In-Distribution)* | | | | | |
> | RGB (SD3.5) | 24×16×16×16 | 16592 | 21869 | 147 | 33.7 |
> | **RGB (SD3.5) + Tracks (Our VAE)** | **24×24×16×16** | **15399** | **20414** | **136** | **28.2** |
> | *Kubric (Out-Of-Distribution)* | | | | | |
> | RGB (SD3.5) | 24×16×16×16 | 13661 | 17004 | 115 | 28.0 |
> | **RGB (SD3.5) + Tracks (Our VAE)** | **24×24×16×16** | **13386** | **16694** | **109** | **24.6** |
>
> The results demonstrate that joint diffusion of RGB and trajectories substantially improves the motion quality extracted from generated RGB frames across all metrics. Notably, the substantial improvement in rigidity proxy metric (LRTL) demonstrates that the generated motion is more rigid and consequently more physically plausible.
>
> We included this evaluation in Section 4.3.

---

> ### Author Response · Authors · 2025-12-03
> **More Real-World Data Results and Visualizations**
>
> To further validate the method using real-world data, we perform the following experiments on the real-world driving dataset CityScapes, involving egocentric camera motion.
>
> **Training data.** We train all models on the CityScapes dataset, which consists of 2,976 clips of 30 frames each. Using CoTracker3, we extract a 56×112 grid of trajectories from each clip for both training and evaluation.
>
> **Implementation.** For computational efficiency, we train our method and Track2Act at a downsampled image resolution of 224×448. WAN1.3B is fine-tuned at its native 480p resolution, but we downsample its generated videos to 224×448 to ensure fair comparison with our method and Track2Act. All methods are evaluated using 28×56 trajectory grids. The complete fine-tuning protocol is detailed in Appendix D.3.
>
> **Evaluation.** We evaluate all methods on 500 unseen Cityscapes validation clips of 30 frames each. Since only one ground truth future is available, we follow the evaluation protocol described in Section 4.1, reporting MSE averaged across three different aggregation methods.
>
> **Results**
>
> | Model            |   MeanT |    Mean |     Min |
> |:-----------------|--------:|--------:|--------:|
> | Track2Act        | 7037.88 | 9305.63 | 4393.58 |
> | WAN (fine-tuned) | 2650.59 | 3495.43 | 1799.04 |
> | **Ours (L)**     | **1475.68** | **1565.06** | **1240.60** |
>
> As shown in the table, our method substantially outperforms both Track2Act and fine-tuned WAN across all metrics. Please also see our qualitative analysis in the Appendix, which shows that our method  produces higher-quality motion overall.
> Interestingly, WAN frequently hallucinates in scenes where the car makes a turn, generating RGB outputs that confuse the point tracker, while Track2Act often fails to account for (distant) objects in the scene, such as pedestrians or cars. These limitations may be a consequence of their limited input image conditioning: Track2Act uses a small pre-trained ResNet to represent the entire scene as a single vector for adaptive normalization. In contrast, we condition on patch-level tokens from DINO through cross-attention in every block, providing richer information about the input image.

---

### Author Response · Authors · 2025-12-03
**Summary**

We thank the reviewers for their positive assessment and constructive feedback, which further improved our work. We have addressed all concerns and responded in detail to the reviewers. We provide a summary here.

**Generalization to real-world scenarios:** Reviewers h2xU and VFf8 raised questions about our method's ability to handle complex, real-world physics. To address this, we trained and evaluated on the **CityScapes** dataset, demonstrating that our approach successfully handles real data with both egocentric camera motion and real-world traffic scenarios from diverse viewpoints. Our results show it outperforms both the fine-tuned pixel-based WAN and the trajectory-based Track2Act methods.

**Multi-frame prediction:** In response to Reviewer VFf8's request, we extended our method to multi-frame prediction. We trained a multi-frame version of our model that successfully estimates velocity from multiple input frames and provided quantitative evaluation with qualitative results in the Appendix.

**Downstream tasks:** In response to Reviewer h2xU's request, we demonstrated that our trajectory generation method can be applied to improve motion quality in video generation.

**Related work:** Following suggestions from Reviewers VFf8 and GBJN, we added further references to relevant prior work in our Related Work section, accompanied by detailed discussion.

**Ablation studies:** Per Reviewer h2xU's request, we conducted an ablation study on query sampling density with comprehensive analysis and discussion.

**Additional qualitative results:** In response to Reviewer GBJN, we expanded the supplementary PDF with two additional pages of qualitative results on Physics101 (Appendix A.6) and included corresponding animations in the offline sample viewer. In response to Reviewer h2xU, we expanded the supplementary PDF with qualitative results on Cityscapes and added approximately 10 interactive animations to the sample viewer.

All improvements and clarifications were incorporated into the paper and supplementary material.

---

### Meta-Review · Area_Chair_Vzor · 2026-01-06

**Summary:**

I recommend accepting this paper. The submission proposes a novel approach to motion forecasting by formulating the task as the conditional generation of dense trajectory grids, rather than direct pixel generation. The method leverages a video-like diffusion model architecture (specifically flow matching) to predict point trajectories, demonstrating advantages in capturing scene-wide dynamics and uncertainty compared to prior regressors and video generators.

The initial reviews were generally positive (scores of 6, 6, and 8), acknowledging the novelty of the trajectory-based approach and the comprehensive evaluation on synthetic datasets. However, reviewers raised valid concerns regarding the model's generalization to real-world data, the limitation of single-image forecasting, and the lack of downstream task evaluations.

The authors provided an exceptionally thorough rebuttal, conducting significant additional experiments, including training on the Cityscapes dataset to prove real-world viability, extending the architecture to handle multi-frame inputs, and performing ablations on sampling density. Given the strength of the core idea and the rigorous effort to address reviewer concerns during the discussion period, I believe this paper makes a valuable contribution to the field of motion generation.

**Reviewer Concerns:**

**Addressed Concerns**

(1) Generalization to Real-World Data (Reviewers h2xU, VFf8): Both reviewers questioned the method's performance outside of synthetic environments like Kubric and Physics. The authors addressed this convincingly by training and evaluating on the Cityscapes dataset, demonstrating that their trajectory-based method outperforms both Track2Act and fine-tuned video generators (WAN) in real-world egocentric driving scenarios.

(2) Single-Image Constraint (Reviewer VFf8): The reviewer questioned why the model was limited to single-image inputs given the ambiguity of motion without velocity. The authors successfully extended their framework to accept multi-frame inputs. They demonstrated that adding frames reduces prediction diversity (uncertainty) and allows the model to infer velocity, effectively turning a perceived limitation into a flexible feature.

(3) Lack of Downstream Tasks (Reviewer h2xU): The reviewer noted a lack of downstream evaluation. The authors implemented a new experiment using their generated trajectories to condition image-to-video generation. They showed that jointly diffusing RGB and trajectories improves the motion quality of the generated video, establishing a clear downstream utility.

(4) Query Point Sampling Density (Reviewer h2xU): The reviewer requested an analysis of sampling density. The authors provided a new ablation study showing that denser grids ($32\times32$) consistently outperform sparser ones, validating their "quasi-dense" design choice.

(5) Missing Citations (Reviewers VFf8, GBJN): The reviewers pointed out missing literature, including Walker et al. (2016) and flow-grounded video prediction work. The authors have incorporated these citations and discussed the distinctions in the revised manuscript.

**Outstanding Concerns**

(1) Robotics Control Policies (Reviewer h2xU): The reviewer suggested evaluating robot manipulation tasks. The authors argued this was out of scope, focusing instead on motion generation quality. While a valid defense, a direct policy learning experiment would have strengthened the paper further. However, this does not outweigh the paper's contributions to generative modeling.


(2) User Study Statistics (Reviewer GBJN): The reviewer felt the 52% preference rate in the user study was "weak". The authors clarified that in a 3-way comparison, 52% is significantly above the random chance baseline of 33%. While the concern is technically addressed, the preference margin is not overwhelming, though sufficient for acceptance.

**Reviewer Scores:**

The reviewers were fairly positive initially, but the extensive additional experiments provided during the rebuttal likely would have improved the scores of the more skeptical reviewers had they confirmed the updates.

(1) Reviewer h2xU (Initial: 6 -> 6/8). The reviewer explicitly listed "Lack of Downstream Task," "No Analysis of Query Point Sampling," and "Insufficient Real-World Data" as weaknesses. The authors directly and successfully addressed all three with new experiments (Cityscapes, Video Gen, Density Ablation).

(2) Reviewer VFf8 (Initial: 6->6/8). The reviewer's primary skepticism regarding the "strange" single-image task setting was addressed by the multi-frame extension, which showed the model can handle video context when available.

(3) Reviewer GBJN (Initial: 8). I anticipate this score would remain an 8. This reviewer was already supportive, and the inclusion of missing citations and clarification on the user study statistics reinforces their positive assessment.

---

### Decision · Program_Chairs · 2026-01-26

Accept (Poster)